# One-step generation of tumor models by base editor multiplexing in adult stem cell-derived organoids

Maarten H. Geurts[1,2,3,8] ✉, Shashank Gandhi[1,5,8], Matteo G. Boretto[1,2,8], Ninouk Akkerman [1,2], Lucca L. M. Derks [2,3], Gijs van Son [1,2,6], Martina Celotti[1,2], Sarina Harshuk-Shabso[1,2], Flavia Peci[2,3], Harry Begthel[1,2], Delilah Hendriks [1,2,3], Paul Schürmann[1,2], Amanda Andersson-Rolf[1,2], Susana M. Chuva de Sousa Lopes[4], Johan H. van Es[1,2], Ruben van Boxtel [2,3] & Hans Clevers [1,2,7] ✉

Optimization of CRISPR/Cas9-mediated genome engineering has resulted in base editors that hold promise for mutation repair and disease modeling. Here, we demonstrate the application of base editors for the generation of complex tumor models in human ASC-derived organoids. First we show efficacy of cytosine and adenine base editors in modeling *CTNNB1* hot-spot mutations in hepatocyte organoids. Next, we use C > T base editors to insert nonsense mutations in *PTEN* in endometrial organoids and demonstrate tumorigenicity even in the heterozygous state. Moreover, drug sensitivity assays on organoids harboring either *PTEN* or *PTEN* and *PIK3CA* mutations reveal the mechanism underlying the initial stages of endometrial tumorigenesis. To further increase the scope of base editing we combine SpCas9 and SaCas9 for simultaneous C > T and A > G editing at individual target sites. Finally, we show that base editor multiplexing allow modeling of colorectal tumorigenesis in a single step by simultaneously transfecting sgRNAs targeting five cancer genes.

Adult stem cell-derived (ASC) organoids can be derived from most epithelial tissues and are emerging as an important tool to bridge the gap between 2D cell lines and in vivo animal models. With their 3D-organization and cellular heterogeneity, organoids closely resemble the tissue of origin and are therefore an attractive model system to study healthy and diseased human tissues[1]. Oncological research may benefit from organoid technology. Patient-derived tumor organoid (a.k.a. tumoroid) biobanks can be established from biopsies, surgical resections[2,3], and even from Pap brushes[4]. These tumoroid biobanks

can then be used as in vitro models of tumorigenesis and offer a significant advantage over conventional 2D cell culture technology[5,6]. Moreover, patient-derived tumoroid models also show promise in testing drug efficacy and potentially determining a personalized treatment for the patient[7–12].

Isogenic models of human cancer can be generated in the lab by introducing tumor-causing mutations in wild-type organoids derived from the tissue of interest. CRISPR/Cas9 has emerged as an efficient strategy for genome engineering and has been successfully used in

[1]Hubrecht Institute, Royal Netherlands Academy of Arts and Sciences (KNAW) and University Medical Center Utrecht, 3584 CT Utrecht, the Netherlands. [2]Oncode Institute, 3521AL Utrecht, the Netherlands. [3]Princess Maxima Center for Pediatric Oncology, 3584 CS Utrecht, the Netherlands. [4]Anatomy and Embryology, Leiden University Medical Center, Einthovenweg, 2333 ZC Leiden, the Netherlands. [5]Present address: Miller Institute for Basic Research in Science, University of California, Berkeley, CA 94720, USA. [6]Present address: Princess Maxima Center for Pediatric Oncology, 3584 CS Utrecht, the Netherlands. [7]Present address: Pharma Research Early Development, Basel, Switzerland. [8]These authors contributed equally: Maarten H. Geurts, Shashank Gandhi, Matteo G. Boretto. ✉e-mail: M.H.Geurts-6@prinsesmaximacentrum.nl; h.clevers@hubrecht.eu

model systems, including organoids, to study tumorigenesis[13–16]. In CRISPR/Cas9-mediated genome engineering, the effector protein Cas9 is guided towards the target site of interest by a single guide RNA (sgRNA) molecule. Cas9 genomic localization is delimited by the protospacer adjacent motif (PAM), which is NGG for SpCas9 while evolved variants of SpCas9 tolerate a broader range of PAM sequences[17–21]. Upon target recognition, the DNA strands are opened in an R-loop and individually cleaved, which results in a double-strand break (DSB)[22,23] Subsequent repair of the DSB by non-homologous end joining (NHEJ) mechanism can result in knock-out of the gene upon gain or loss of base pairs at the break site. While exogenous DNA can be inserted at the target site through homology-directed repair (HDR) mechanism, successful insertion of exogenous DNA is relatively rare and is likely accompanied by loss of the second allele[24]. Unsurprisingly, there are downsides to genome engineering via DSBs. The repair of DSBs introduced by Cas9 has been shown to be error prone and may result in complex rearrangements, including chromothripsis, at the site of repair[25–27].

To circumvent these issues, strategies that avoid DSBs have been recently developed. Fusion of a cytidine deaminase (rAPOBEC1) to a partially inactive nickase-Cas9(D10A) protein results in a cytidine base editor (CBE) that allows for C > T base changes via a uracil intermediate[28] (Supplementary Fig. 1a). Fusing adenine deaminase (an evolved TadA heterodimer) to a nickase-Cas9 creates an adenine base editor (ABE) that mediates A > G base changes via an inosine intermediate[29] (Supplementary Fig. 1b). The deaminases fused to Cas9 in base editing constructs act on single-stranded DNA. Thus, base editors act only within a small window of the single-stranded R-loop that is generated upon Cas9 target recognition[30]. This limited editing window roughly the size of four nucleotides between positions 4 and 8 from the 5' end of the protospacer dictates the need for specific localization of the CRISPR/Cas9 complex. As base editors mediate genetic changes without the need for deleterious DSBs, they represent a promising genome engineering tool for clinical applications. We have previously described the use of ABEs in Cystic Fibrosis patient-derived organoids to functionally repair mutations in CFTR without genome-wide off-targets[31].

In spite of these advancements in the field of genome engineering, generation of cancer models that faithfully recapitulate the genetics of human cancer has been quite challenging. While we and others have previously shown that CRISPR/Cas9-mediated genome engineering can be applied to organoids to study tumorigenesis, modeling of complex genotypes involving mutations at several genomic loci has required sequential introduction of mutations. This sequential method was laborious, resulting in several months' worth of selections and expansion to obtain stable lines that harbored all mutations of interest. Together with the unspecific editing outcomes of conventional Cas9 proteins, this was a major limitation holding the field back. In this paper, we describe an efficient one-step strategy to create complex combinations of the exact single base changes as associated with cancers in human ASC-derived organoids. When combined with functional selection, our method proves to be efficient in establishing a library of mutant organoids with complex genotypes, which we validated by Sanger sequencing. We further demonstrate the versatility and flexibility of multiplexed CRISPR base-editing for cancer modeling across several epithelial tissues by mutating liver, endometrium, and organoids[32–34]. We envision that our approach can be readily adapted to create in vitro models for tumorigenesis of solid human tumors for a vast number of tissues.

## Results

### ABEs and CBEs are applicable for oncogene activation in organoids

Mutational activation of the WNT pathway is observed in multiple cancer types which often involve specific point mutations in the Wnt effector gene, beta-catenin (CTNNB1) (Fig. 1a)[35]. These mutations target a region in the N-terminus of the protein (encoded by exon 3) which is involved in its rapid degradation in the absence of external Wnt signals. Normally, a serine residue at position 45 (S45) is phosphorylated by the priming kinase CK1. This is followed by a GSK3-mediated phosphorylation cascade that starts at threonine-41 (T41) and the subsequent phosphorylation of serines at positions 33 and 37 (S33 and S37) (Fig. 1a). Phosphorylation of S33 and S37 generates a phospho-degron motif (residue 32 to 37), which leads to proteasomal degradation of CTNNB1 upon recognition by the E3-ligase β-TRCP[35].

Previously, Zafra and colleagues have utilized CBEs to mutate the S45 residue, effectively modeling the first step of the phosphorylation cascade[36]. However, modeling the subsequent phosphorylation defects, required an approach that did not rely on conventional Cas9 proteins due to limited PAM sites in the exon 3 locus. We therefore decided to first tackle this problem by exploring alternative SpCas9-ABE variants that recognize non-canonical PAMs. To this end, we turned to fetal hepatocyte organoids and designed three sgRNAs to introduce oncogenic mutations in the CTNNB1 locus Different combinations of these three sgRNAs with base editors enabled us to introduce four unique mutations in the third exon of CTNNB1 (Fig. 1b).

To further ensure that we can faithfully establish clonal organoid lines harboring these mutations, we used a transposon-based integration approach to introduce an antibiotic resistance gene to enable efficient selection of transfected organoids. This involved the co-electroporation of two additional plasmids, one encoding the piggy-bac transposase and another encoding the hygromycin resistance cassette, together with an sgRNA combined with a base editor[37]. We first validated this approach by introducing the recurring W53* mutation in the tumor suppressor gene, TP53. TP53 mutant organoids can be functionally selected for by addition of the compound Nutlin-3 to the selection medium[13,14], thereby providing an elegant platform to test the efficacy of hygromycin selection in picking TP53^W53* mutant organoids (Supplementary Fig. 2a). We performed the electroporation and developed organoids for one week before adding Nutlin-3 in the organoid medium. In parallel, we selected organoids based on their resistance to hygromycin in the medium. We directly compared the abundance of the TP53^W53* mutation in functionally selected organoids with organoids that were either untransfected or electroporated but not functionally selected. We posited that if the hygromycin selection allowed for the positive selection of TP53-mutant organoids, then the TP53^W53* mutation will be well represented in the population. We therefore performed Sanger sequencing on the bulk culture that was untransfected, unselected, hygromycin selected, and nutlin-3 selected and found that while100% of the nutlin-3 resistant bulk culture had the TP53^W53* mutation incorporated (Supplementary Fig. 2b, c), 72% of the hygromycin-resistant clones harbored the mutation (Supplementary Fig. 2b, c)[38]. Taken together, these results indicate that the hygromycin resistance system can be utilized to increase the percentage of mutated organoids in our cultures and enable quantification of the editing efficiency associated with different CRISPR technologies.

Subsequently, we designed sgRNA's that, in combination with CBE or ABE, would allow for the induction of mutations in CTNNB1 that block E3-ligase-dependent degradation. These sgRNA's either target phosphorylation sites (T45, T41, and S33) or target the phosphor-degron site (D32). by co-transfecting ABEs or CBEs with sgRNAs and two plasmids that enable stable integration of a hygromycin cassette[37], we aimed to assess editing efficiencies of ABEs and CBEs (Fig. 1c, Supplementary Fig. 3a) through Sanger sequencing of hygromycin-resistant clones. We first focused on impairing the CK1-mediated phosphorylation of CTNNB1 by mutating the S45 residue to a Proline (hereby referred to as the S45P mutation). Co-transfection of pCMV-SpCas9-ABEmax with sgRNA-1 resulted in 9 out of 23 clones that harbored a homozygous S45P mutation (~40%) (Fig. 1d). An additional silent (P45P) mutation co-occurred in all edited clones (c.135T > C)

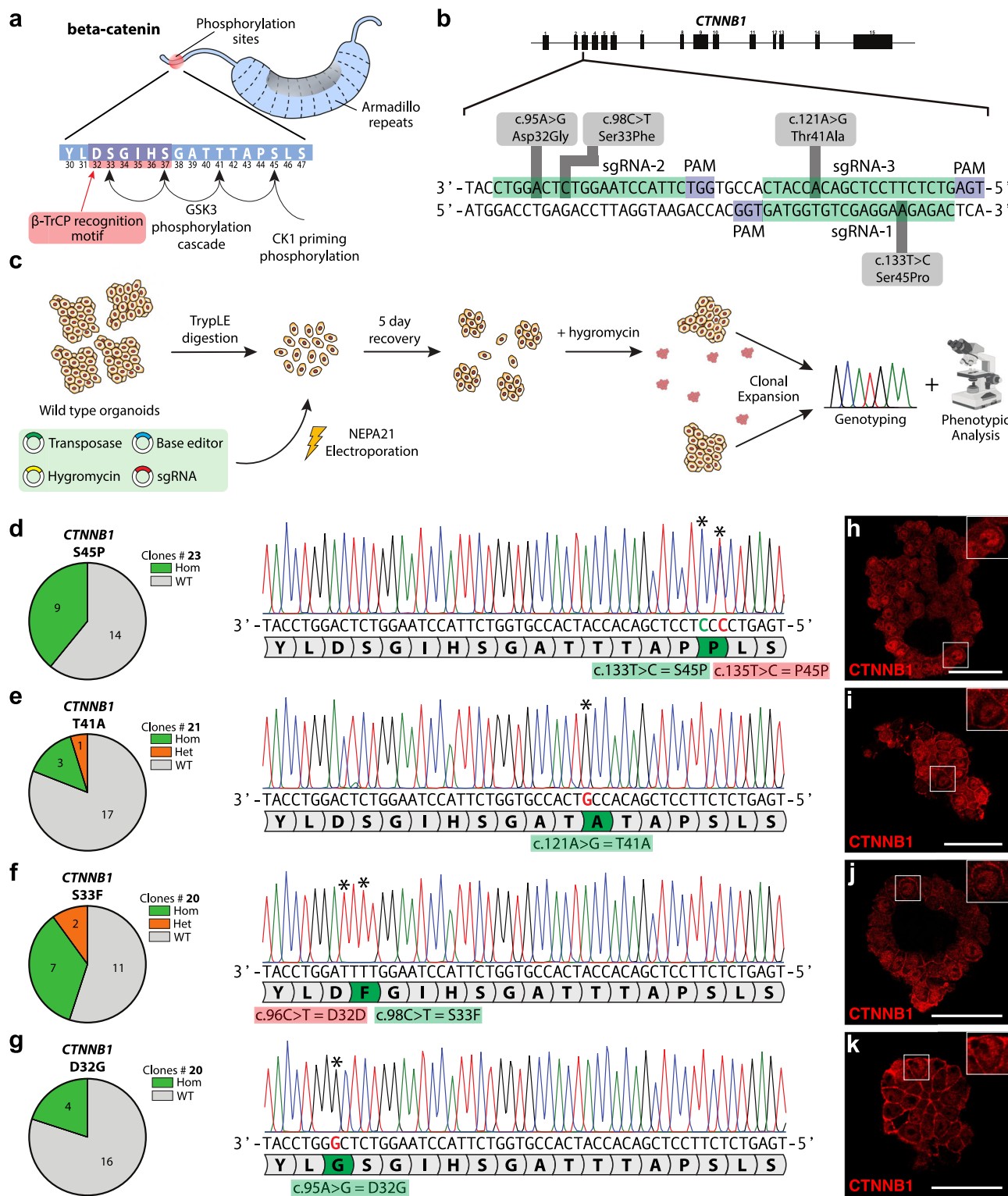

**Fig. 1 | Introduction of oncogenic activation mutations in *CTNNB1* in hepatocyte organoids using conventional and evolved CBE and ABE. a** Protein encoded by the gene *CTNNB1* harbors hot-spot mutations in the N-terminus, which correlate with principles of CTNNB1 degradation by kinases CK1 and GSK3 followed by proteasomal destruction by E3-ligase β-TrCP. **b** Design of 3 sgRNA's for introduction of hot-spot mutations in exon 3 of *CTNNB1*. sgRNA-1 can be used with SpCas9-ABE to introduce priming kinase mutation S45P, sgRNA-2 can be used with SpCas9-ABE to introduce D32G or with SpCas9-CBE to introduce S33F and sgRNA-3 in combination with SpCas9-NG-ABE introduces T41A. sgRNA sequences are in green, PAM sequences are in blue. **c** Principles of hepatocyte organoid electroporation for *CTNNB1* mutagenesis. **d–g** Sanger trace and quantification of editing efficiency by Sanger sequencing of hygromycin-resistant organoids harboring *CTNNB1^{S45P}* (**d**), *CTNNB1^{T41A}* (**e**), *CTNNB1^{S33F}* (**f**), and *CTNNB1^{-D32G}* (**g**). Asterisks highlight mutated nucleotides. Correct mutations are highlighted in green and synonymous but unintended mutations are highlighted in red. **h–k** CBE and ABE mutated hepatocyte organoids on residue S45 (**h**), T41 (**i**), S33 (**j**), and D32 (**k**) show increased intracellular localization of CTNNB1. Scale bar in (**h–k**) is 50 μm. Source data are provided as a Source data file.

(Fig. 1d). Next, we impaired the GSK3-mediated phosphorylation cascade by targeting the T41, S33, and D32 residues. Co-transfection of an evolved Cas9 variant pCMV-SpCas9-NG with sgRNA-2 identified 3 homozygous (-14%) and 1 heterozygous (-5%) T41A mutation out of 21 clones (Fig. 1e). Co-transfection of pCMV-SpCas9-ancBE4max with sgRNA-3 resulted in homozygous S33F mutations in 4 out of 20 clones (20%) (Fig. 1f). The S33F clones all harbored an additional induced point mutation, albeit silent (D32D), two bases upstream (c.96C > T) (Fig. 1g). Finally, co-transfection of sgRNA-2 with pCMV-SpCas9-ABEmax resulted in 2 heterozygous (10%) and 7 homozygous (35%) D32G mutations out of 20 clones (Fig. 1g).

Next, we investigated the impact of these hot-spot mutations on the CTNNB1 protein in mutant organoids. To do this, we immunohistochemically labeled CTNNB1 and employed confocal microscopy to assess its intracellular localization within single cells of clonal wild-type and mutant organoids. We reasoned that CTNNB1 harboring the four point mutations discussed earlier would not be transported to the plasma membrane by the destruction complex comprising of the proteins AXIN, APC, GSK3, CK1, and PP2A for degradation, but would instead be stabilized and translocated to the nucleus to activate downstream Wnt-responsive genes. Indeed, *CTNNB1*-mutant organoids (Fig. 1h–k) showed increased expression in the nucleus, suggesting that beta catenin was translocated to the nucleus for activation of downstream target genes. These data successfully demonstrate the application of ABEs and CBEs for oncogene activation in hepatocyte organoids.

## ABEs and CBEs allow investigation of tumorigenic potential of individual hotspot mutations

According to the Catalog of Somatic Mutations in Cancer (COSMIC), mutations observed across cancer patients at the CK1 priming residue (S45P) are almost twice as abundant compared to the E3 ubiquitin ligase residue (D32) (Fig. 2a). This was in contrast to the increased beta-catenin immunofluorescence we observed in the nucleus (Fig. 2h–k) compared to the wildtype organoids (Fig. 2b), where both S45P and D32G mutations resulted in similar increase in nuclear localization of beta-catenin. To gain a better understanding of the relative intracellular distribution of CTNNB1 in a given cell, we quantified the signal intensity through the plasma membrane and the rest of the cell (combined cytoplasm and nucleus). As expected, CTNNB1 was mainly localized at the plasma membrane in wild-type organoids, presumably bound to adherens junctions[36], whereas intracellular levels in the cytoplasm and nucleus was low (Fig. 2d, Supplementary Fig. 3b). In contrast, organoids harboring the homozygous S45P mutation showed reduced levels of CTNNB1 at the plasma membrane and increased amounts in the cytoplasm and nucleus (Fig. 2e, Supplementary Fig. 3c, d), consistent with an "overactivation" phenotype. Similarly, mutation in the E3-ligase recognition domain (D32G) showed increased intracellular localization (Fig. 2f, Supplementary Fig. 3e, f). Next, as a readout for activated Wnt signaling in mutant organoids, we calculated the ratio of mean CTNNB1 intensities in the cytoplasm/nucleus and the plasma membrane. Relative to the wild-type control, both point mutations resulted in increased ratios (Fig. 2g), and by extension, activation of Wnt signaling. Interestingly, the ratio of intracellular to membrane intensity better recapitulated the observed frequency of mutations at these two loci. This suggested that a higher proportion of activated beta-catenin is available for translocation into the nucleus relative to the membrane following the S45P mutation as compared to D32G, where a significant proportion still gets sequestered to the membrane.

To further characterize the effect of these point mutations on Wnt pathway activation, we cultured mutant organoids on a medium that lacked pathway activators R-spondin and the GSK3β inhibitor CHIR, components that are otherwise used in the expansion medium. We first confirmed that the mutant organoids successfully grow in medium lacking R-spondin and CHIR. All four mutant organoids described in Fig. 2, but not the wild type, exhibited sustained growth in Wnt/Rspondin-independent conditions for at least five passages (after which the experiment was terminated) (Supplementary Fig. 4a, b). Next, we harvested RNA from both S45P and D32G mutant organoids 24 h following Wnt-removal and performed bulk RNA-sequencing. We directly compared the number of differentially expressed genes in these mutants with wild-type organoids grown without Wnt/Rspondin. RNA-seq analysis revealed considerable overlap between the three conditions, with 334 genes significantly enriched and shared between the three genotypes. Interestingly, both S45P and D32G mutants showed overlap of 668 genes, whereas similar number of genes were uniquely enriched in the two backgrounds (296 for D32G and 298 for S45P). Consistent with CTNNB1 intensity measurements, Wnt-pathway target genes were upregulated compared to wild-type organoids, indicative of cell-autonomous WNT pathway activation, results that were further corroborated by qPCR on four common Wnt-pathway target genes, including *AXIN2, DKK1, LGR5,* and *RNF43* (Supplementary Fig. 4c)[39,40]. Furthermore, both S45P and D32G showed upregulation of Notum, a negative Wnt regulator previously shown to promote cell competition in APC-deficient colorectal cancer cells[41]. Interestingly, Reg3a was also upregulated in both mutants, and high levels of Reg3a were previously shown to promote proliferation and tumorigenesis of colorectal and hepatocellular carcinomas[42,43]. To better understand the difference between the impact of S45P and D32G mutations on transcription of downstream genes, we directly compared the two genotypes and found that while 178 genes were differentially enriched in the D32G mutant, 185 genes were differentially enriched in the S45P background. Several of these genes were transcription factors (Fig. 2j) and known Wnt target genes (Fig. 2k), including *ATF3, LEF1,* and *HNF4A.* Further analysis of genes specifically upregulated in the S45P mutant revealed REC8 as the most upregulated gene. REC8, a cohesion that functions during meiosis, was previously linked to colorectal cancer and hepatocellular carcinoma by promoting invasiveness and metastasis[44]. Similarly, RALYL, an RNA-binding protein that functions as a transcriptional regulator, mainly by regulating the stability of mRNA transcripts, was also specifically upregulated in S45P. Prior work showed that RALYL stabilizes Tgfb2 transcripts in hepatocellular carcinoma, thus increasing the stemness of HCC cells[45]. Gene Ontology analysis on the list of enriched genes in the D32G background revealed a significant upregulation of the "plasma membrane" cellular compartment. This included genes such as Cadherin 17 (Fig. 2l), whose targeted inactivation results in reduced Wnt pathway activation. These results contextualized our immunohistochemistry data, where more beta-catenin was present on the membrane (Fig. 2f) in the D32G background. (Fig. 2i). Taken together, these results confirm that hot-spot mutations in the third exon of *CTNNB1* lead to significant Wnt pathway activation, and that both conventional and evolved forms of Cas9 can be used to introduce point mutations in human ASC-derived organoids by ABEs and CBEs.

## CBE-mediated CRISPR-stop can be used in organoids to model tumorigenesis in endometrial organoids

CBEs, as part of a technique called "CRISPR-stop", have been used to introduce stop codons at Glutamine (Q), arginine (R), and Tryptophan (W) residues[46], thereby allowing inactivation of tumor suppressor genes in human ASC-derived organoids (Supplementary Fig. 1c). Endometrial cancer (EC) is the fourth most common tumor type among women (Siegel et al.[47]). ECs are classified as endometrioid and non-endometrioid tumors, with the former accounting for more than 80% of the cases. Phosphatase and tensin homolog *(PTEN)* is the most frequently mutated gene in ECs and harbors the highest relative number of mutations across tumor types according to COSMIC (Fig. 3a). *PTEN* mutations are often observed early during tumorigenesis, whereas a complete loss is considered to be a late event[48]. The

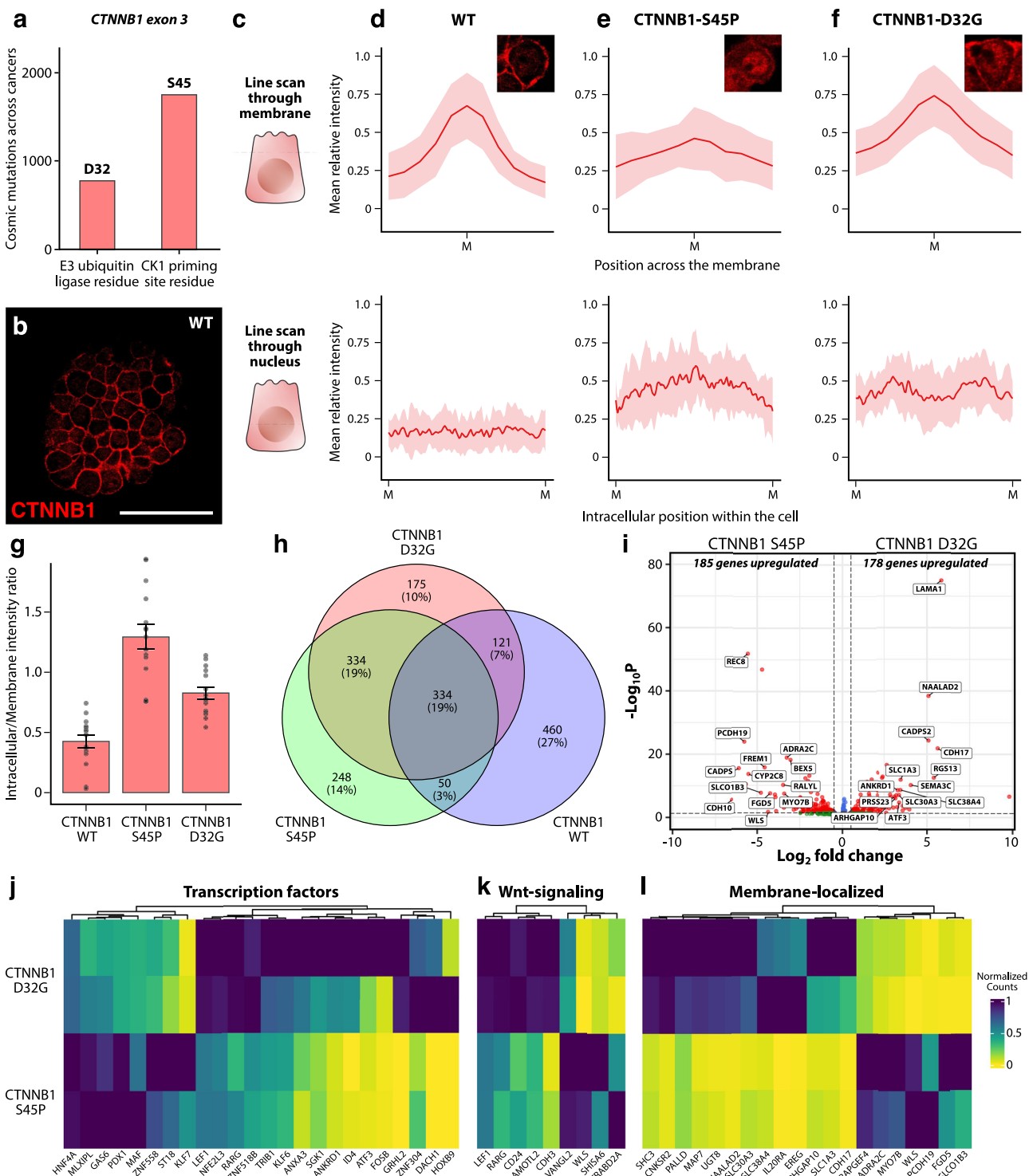

**Fig. 2 | Impaired localization of CTNNB1 upon hot-spot mutation induction by CBE and ABE. a** Amount of hot-spot mutations as quantified by the COSMIC database for somatic mutations in cancer. The y-axis shows number of mutations across all tumor types in the database. X-axis shows Amino acid residue location. **b** CTNNB1 is localized to the plasma membrane in wild-type organoids, while the intracellular levels are low. **c** Illustration of the line scans through the plasma membrane and nucleus to quantify CTNNB1 immunofluorescence intensity. **d**–**f** Quantification of mean relative intensity of CTNNB1 through the plasma membrane and cytoplasm/nucleus across wildtype (**d**), S45P (**e**), and D32G (**f**) genotypes (inset: representative cells for each genotype). Solid line represents the mean fluorescence intensity. Error band represents standard error of the mean

(s.e.m.). **g** The CTNNB1 intensity ratio of intracellular to plasma membrane is significantly different across different mutants when compared to the wild-type control ($n = 15$ cells analyzed over two independent organoid clones). Bar plots are presented as mean values. Error bars represent standard error of the mean (s.e.m.). **h** Three-way venn diagram showing overlap of significantly enriched genes between wild type, S45P, and D32G genotypes. **i** Volcano plot showing genes that were differentially expressed between the S45P and D32G genotypes. Genes that have an absolute fold change of greater than 3 have been labeled. **j**–**l** Heatmaps showing transcription factors (**j**), Wnt-target genes (**k**), and the 20-most enriched membrane-bound genes (**l**) selected from all significantly enriched genes extracted from the volcano plot. Source data are provided as a Source data file.

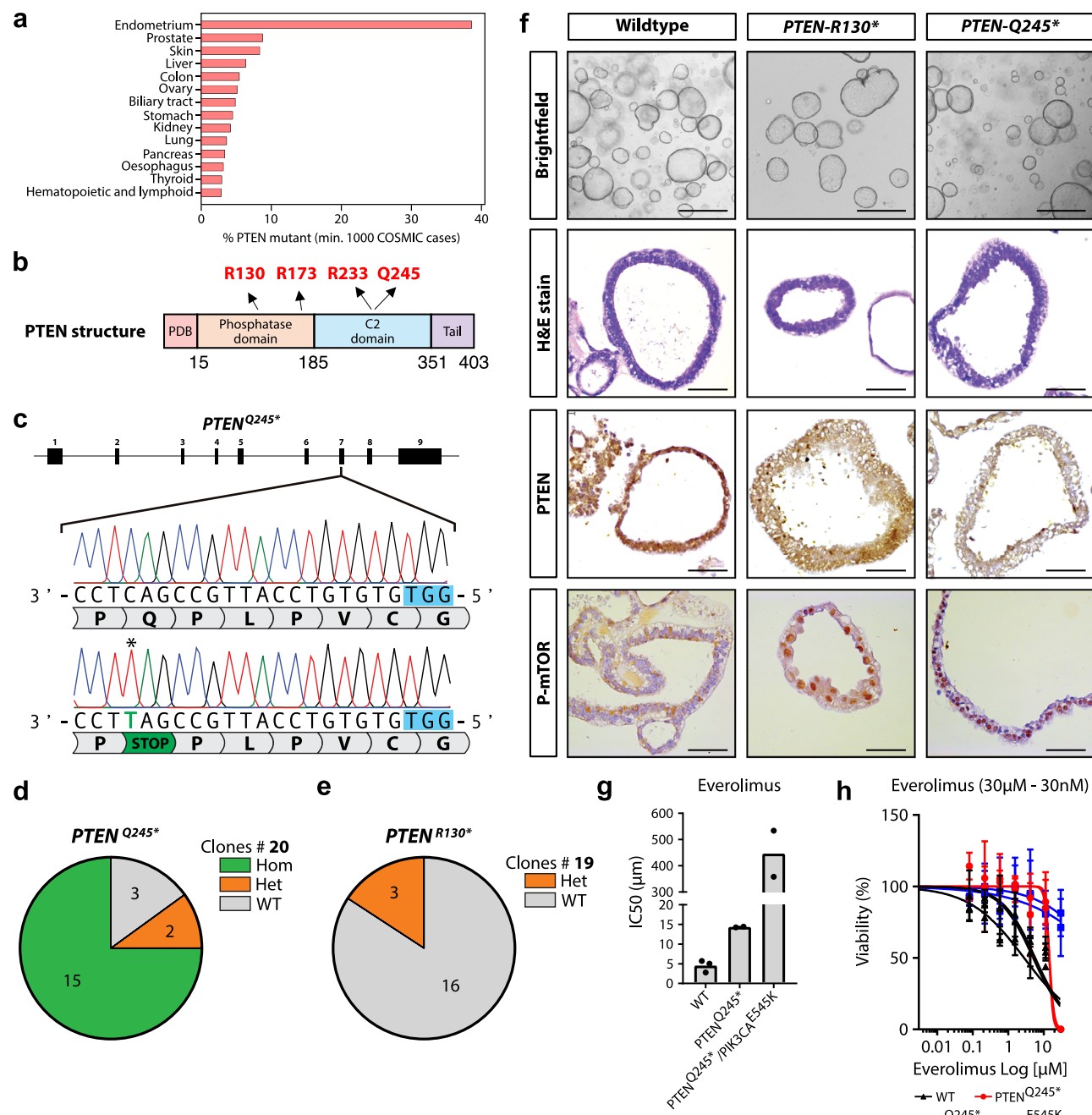

**Fig. 3 | CBE CRISPR-STOP effectively introduces nonsense mutations in *PTEN* in endometrial organoids. a** Bar graph displaying the percentage of *PTEN* mutations in various cancer types ordered according to frequency. The endometrium harbors the highest percentage of *PTEN* mutations. **b** Structure of the PTEN protein showing the different domains. Mutational hotspots are highlighted in red. **c** Sequence alignment of the *PTEN*$^{Q245}$ locus showing successful C > T transition. The PAM sequence is indicated in blue. The asterisk highlights the edited residue. **d** Pie chart reporting the efficiency of the *PTEN*$^{Q245*}$ sgRNA. The number of clones per genotype is indicated. **e** Pie chart reporting the efficiency of the *PTEN*$^{R130*}$ sgRNA. The number of clones per genotype is indicated. **f** Panel showing representative pictures of *PTEN* WT and mutant organoids. *PTEN* nonsense mutations result in lower immunoexpression of PTEN protein and increased phosphorylated mTOR expression. Scale bar 500 µm for brightfield and 50 µm for histology. **g** Dose−response curve reporting the sensitivity to Everolimus (mTOR inhibitor) of WT and PTEN, PTEN/PIK3CA mutant organoids. The viability is indicated on the Y-axis while the inhibitor concentration is indicated on the X-axis in logarithmic scale. Both mutants show reduced sensitivity. Data is derived from three technical replicates. **h** Bar graph representing the IC50 of Everolimus in different organoid lines, n = 3 (WT), n = 2 *PTEN*$^{Q245*}$ and *PTEN*$^{Q245*}$/*PIK3CA*$^{E545K}$. Statistics derived from three technical replicates. Data are presented as mean values +/− SD. Source data are provided as a Source data file.

PTEN protein contains mutational hotspots within two major domains that mediate its phosphatase activity and plasma membrane localization, respectively (Fig. 3b).

We first used SpCas9-CBE to target the Q245 residue in exon 7 of *PTEN*. Similar to our experimental setup in hepatocyte organoids, we used stable integration of a hygromycin cassette to select electroporated organoids, which were then clonally expanded and genotyped using Sanger sequencing. Robust GFP expression through the CBE plasmid was detected at 24 h after electroporation (Supplementary Fig. 5a, b). Co-transfection of pCMV-AncBE4max-P2A-GFP and the sgRNA resulted in Q245* homozygous mutations in 15 out of 20 clones (75%) (Fig. 4c, d). We occasionally observed non-synonymous

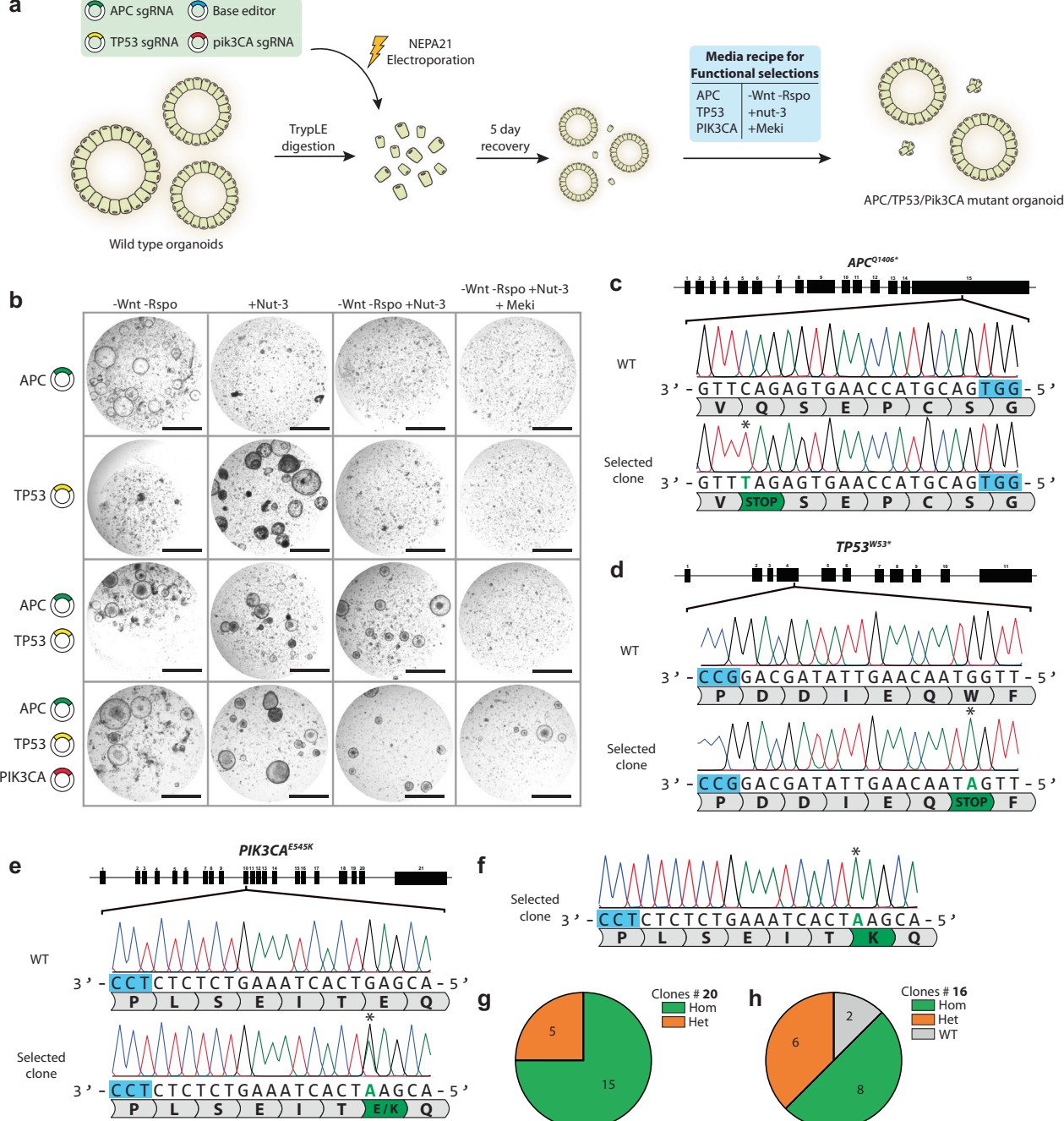

**Fig. 4 | CBE multiplexing allows for simultaneous tumor suppressor inactivation and oncogene activation. a** Principles of intestinal organoid electroporation and subsequent functional selection by growth factor manipulation. **b** Brightfield images of organoids selected for *TP53*, *APC*, and *PIK3CA* mutations. sgRNA combinations are shown on the left. **c** Sanger sequencing of APC Q1406* mutant clones that survive Wnt-pathway selection. Asterisk highlights mutation of interest and PAM is shown in blue. Scale bars are 2000 μm. **d** Sanger sequencing of *TP53^W53*^* mutant clones that survive Nutlin-3 selection. Asterisk highlights mutation of interest and PAM is shown in blue. **e** Sanger sequencing of heterozygous *PIK3CA^E545K^* mutant clones that survive Meki selection. Asterisk highlights mutation of interest and PAM is shown in blue. **f** Sanger trace of homozygous *PIK3CA^E545K^* mutant clones that survive Meki selection. Asterisk highlights mutation of interest and PAM is shown in blue. **g** Quantification of mutations found in clones that are selected on Wnt-pathway withdrawal, Nutlin-3, and Meki addition. **h** Quantification of editing efficiency of the *PIK3CA^E545K^* guide by Sanger sequencing of 16 clones surviving only APC and TP53 selections. Source data are provided as a Source data file.

mutations in the 2nd, 7th, and 8th cytosine of the editing window, likely due to the presence of poly-C repeats. Interestingly, our initial attempt of introducing point mutations at the R130 hotspot residue using SpCas9-NG-CBE was unsuccessful with two different sgRNAs (Supplementary Fig. 5c, d). It has been previously shown that the editing efficiency of both ABEs and CBEs relies on factors such as the editing position within the protospacer and the tri-nucleotide sequence around the editing site[49]. In particular, lower efficiency can

be observed at positions 3, 4, and 8 of the protospacer sequence. The trinucleotide ACG around the editing site can also hamper editing efficiency. Both our initial sgRNAs targeted a cytosine at position 3 (sgRNA-1) or 8 (sgRNA-2) of the protospacer sequence within the ACG trinucleotide which likely explains the low editing efficiency (Supplementary Fig. 5c). To circumvent this drawback, we employed SpRY-CBE, a base editor variant which recognizes an NAN motif adjacent to the protospacer[21]. This allowed us to introduce R130* heterozygous

mutations in 3 out of 19 clones (~11%) (Fig. 3e, Supplementary Fig. 5c–e), thereby overcoming a major bottleneck to successfully model early endometrial tumorigenesis[50].

Next, we assessed the impact of $PTEN^{Q245*}$ and $PTEN^{R130*}$ on endometrial organoids. Histological examination of both mutants revealed that the organoids retained their cystic morphology similar to WT organoids (Fig. 3f). We confirmed a downregulation of PTEN protein expression in *PTEN* mutant organoids (Fig. 3f) and increased expression of the downstream effector of the PTEN/PI3K pathway mTOR, whose increased phosphorylation was detected in the nucleus of mutant cells (Fig. 3f). These data indicate that even heterozygous loss-of-function mutations can result in PTEN/PI3K pathway overactivation. While this hypothesis has been previously proposed[48,50–52], it had never before been tested in an isogenic model. However, our toolkit allowed the generation of this mutation model and subsequently confirm the activation of the PTEN/PI3K pathway in a heterozygous background. Endometrioid endometrial cancers are often burdened by multiple mutations along the PI3K pathway. Indeed, the combined presence of *PTEN* and *PIK3CA* or *PIK3R1* mutations are frequently observed suggesting that both loss of tumor suppressor and activation of oncogenes is required during tumor progression[53]. To further refine our endometrial tumorigenesis model, we next introduced the $PIK3CA^{E545K}$ mutation in the $PTEN^{Q245*}$ mutant organoids with the use of the same base editor: pCMV-AncBE4max-P2A-GFP. We expanded these organoids for one week in complete medium and looked for transcriptional changes using bulk RNA-sequencing. Comparison of enriched genes in the double-mutant organoids with wildtype organoids revealed a list of PI3K pathway target genes (Supplementary Fig. 6a). As expected, we observed a downregulation in *PTEN* gene expression, which is likely a result of the premature stop codon we introduced in the organoids (Supplementary Fig. 6b). We also observed an upregulation of the genes *AKT, mTOR, PIK3R3, IGFBP5* in mutant organoids. We next turned to Gene Set Enrichment Analysis (GSEA) to capture global transcriptomic changes, which resulted in a correlation with mTORC1 signaling in our mutant organoids (Supplementary Fig. 6b).

To further assess the impact of *PTEN/PIK3CA* mutations on endometrial organoids, we performed a drug sensitivity assay targeting the PTEN/PI3K pathway at multiple levels using the homozygous $PTEN^{Q245*}$ and $PTEN^{Q245*}/PIK3CA^{E545K}$ organoids. In contrast to wild-type organoids, we observed decreased sensitivity to the mTOR inhibitor Everolimus in PTEN/PIK3CA mutants, likely due to increased mTOR signaling output in mutant organoids (Fig. 3g, h). This was further supported by the effect on cell viability with the pan-FGFR inhibitor AZD-4547 (Supplementary Fig. 6c, d) and the PIK3CA inhibitor Alpelisib (Supplementary Fig. 6f). Taken together, these results demonstrate the significant advantage the toolkit we have developed in this study has to generate isogenic models of novel mutations in early tumorigenesis[48,50–52].

## CBE multiplexing allows for simultaneous oncogene activation and tumor suppression inactivation

Thus far, we have established conventional and evolved variants of Cas9-CBEs and -ABEs as tools to generate organoid lines that harbor either mutations that activate oncogenes or inactivate tumor suppressors. However, to demonstrate their combinatorial efficacy, we turned to colorectal cancer tumorigenesis modeling.

We first separately optimized the CRISPR-stop strategy for disruption of genes that are commonly mutated in colorectal cancer. Loss-of-function mutations in *APC* are often the first to occur during colorectal tumorigenesis. Mutations in APC mostly occur in exon 15 and cause Wnt-pathway independence[54]. Since wild-type organoids heavily rely on Wnt-3A in the expansion medium for sustained growth, APC mutants can be readily selected by removing Wnt from the "selection" medium[13,14]. We designed two sgRNAs that, when combined with CBEs, introduce the recurring mutations Q1406* and R1114* in the

*APC* locus. pCMV-AncBE4Max-P2A-GFP was co-transfected with both sgRNAs, and after a recovery period of 5 days in expansion medium, R-spondin1 and Wnt-3A were removed (Fig. 4a). After 14 days on this selection medium, we observed organoids growing without Wnt, whereas organoids transfected with control scrambled sgRNA failed to grow (Fig. 4b, Supplementary Fig. 7a). Sanger sequencing of organoids growing without Wnt revealed correct mutation introduction using both the $APC^{Q1406*}$ and $APC^{R1114*}$ sgRNAs (Supplementary Figs. 7b, 3c). However, the outgrowth on No-Wnt selection medium was higher for the $APC^{Q1406*}$ sgRNA, and we therefore used this construct for subsequent multiplexing experiments. Another gene that is commonly mutated in colorectal cancer together with *APC* is the tumor suppressor protein TP53. We performed a similar assay as described above to identify a potent TP53 sgRNA. As described earlier in this study, *TP53* mutants can be functionally selected by addition of the compound Nutlin-3 to the selection medium[13,14]. We designed two sgRNAs that, combined with CBEs, could introduce the recurring mutations R213* and W53* in *TP53*. We co-transfected pCMV-AncBE4Max with both sgRNAs and recovered electroporated cells for 5 days in expansion medium. On the 6th day, we added Nutlin-3 to the selection medium (Fig. 4a) and maintained the organoids in culture for two weeks. While organoids transfected with the scrambled sgRNA control failed to grow in the presence of Nutlin-3, we observed several mutant organoids electroporated with the R213* and W53* sgRNAs growing in the presence of Nutlin-3 (Fig. 4b, Supplementary Fig. 7d). Sanger sequencing of the Nutlin-3 resistant clones transfected with the R213* sgRNA revealed correct R213* induction, albeit occasionally accompanied by the mutation T211I (Supplementary Fig. 7e). On the other hand, Nutlin-3 resistant clones transfected with the W53* sgRNA revealed correct mutation on the W53 residue without off-target mutations. Therefore, we selected this sgRNA for subsequent multiplexing experiments.

We then co-transfected pCMV-AncBE4Max-P2A-GFP with both $APC^{Q1406*}$ and $TP53^{W53*}$ and selected organoids by removal of Wnt-3A/R-spondin and addition of Nutlin-3 (Fig. 5b). Clonal expansion followed by Sanger sequencing revealed the expected stop codon mutations in both *APC* and *TP53* (Fig. 4c, d).

Another gene that is commonly mutated in colorectal cancer together with APC and TP53 is PIK3CA, which harbors a hotspot mutation E545K in its tenth exon. Activating mutations in PIK3CA can be selected for by addition of Mek inhibitors (Meki)[14] to the selection medium. We used the previously described PIK3CA^E545K sgRNA to recreate this oncogenic mutation in intestinal organoids, and co-transfected it with pCMV-AncBE4Max, and the $APC^{Q1406*}$ and $TP53^{W53*}$ sgRNAs. Following a 5-day recovery on expansion medium, we subjected the electroporated cells to a selection medium that lacked Wnt-3A/Rspondin-1 and contained Nutlin-3 and Meki. We observed outgrowth 14 days after the start of selection (Fig. 4b). Clonal expansion followed by Sanger sequencing of 20 clones revealed 5 heterozygous (25%) and 15 homozygous (75%) E545K mutation, for a total of 100% mutation introduction (Fig. 4e–g). We assessed the efficiency of the E545K guide by functionally selecting for mutations in APC (removal of Wnt) and TP53 (addition of Nutlin-3) but genotyped for the PIK3CA^E545K mutation. This analysis revealed 8 homozygous (50%) and 6 heterozygous (37.5%) mutations out of 16 clones for a total editing efficiency of 87.5% (Fig. 4h).

## Cas9 homolog multiplexing allows for simultaneous CBE and ABE on distinct target sites

While Cas9 CBEs and ABEs have been individually used to introduce oncogenic activating/tumor suppressor inactivating mutations, their concomitant application has not yet been explored in tumor modeling. This is partly because sgRNAs designed for C > T tumor suppressor inactivation can also be used by SpCas9-ABE to introduce A > G mutations at the same location and vice versa (Fig. 5a). Moreover, not all oncogenic activating mutations are C > T and can therefore be

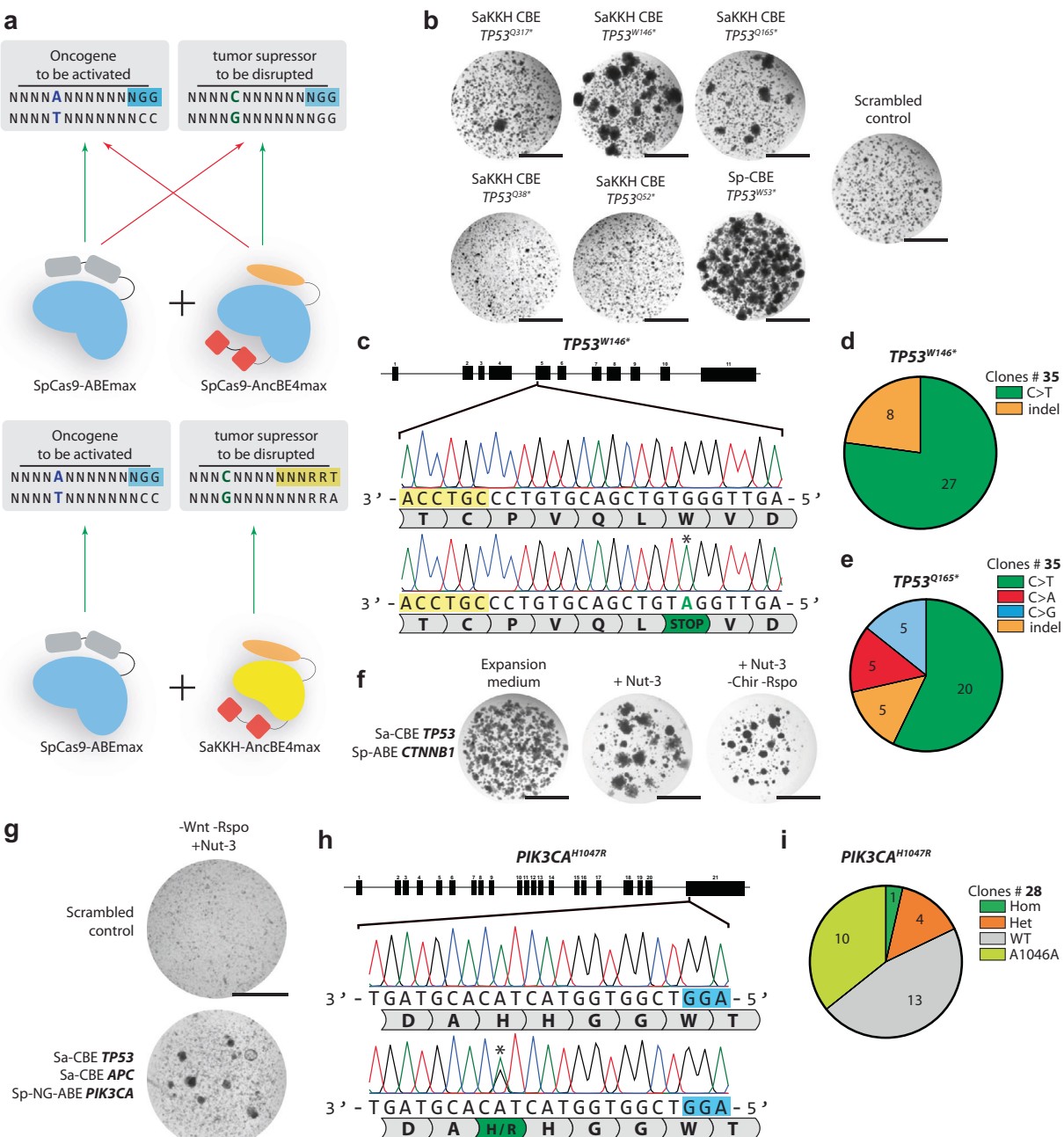

**Fig. 5 | Simultaneous C > T and A > G engineering by Cas9 homolog multiplexing. a** Schematic overview of the problem arising when CBE and ABE is applied using the same cas9-homolog. This can be circumvented by using Sa-CBE together with Sp-ABE. **b** Brightfield images showing hepatocyte organoid outgrowth upon Nutlin-3 selection on 3 SaKKH-sgRNA's compared to a scrambled control sgRNA and our optimal *TP53^W53*^* sgRNA in combination with SpCas9-CBE. Scale bars are 2000 μm. **c** Sanger sequencing of hepatocyte organoids surviving Nutlin-3 selection upon transfection of SaKKH-CBE and a *TP53^W146*^* sgRNA. Edited residue is highlighted with an asterisk and SaKKH PAM is shown in yellow. **d** Quantification of editing observed in Nutlin-3 resistant clones highlights increased indel induction by SaKKH-CBE. Clones with homozygous C > T mutations are shown in green and clones with indels on one of their alleles are shown in orange. **e** Quantification of editing observed in Nutlin-3 resistant clones highlights increased amount of unintended edits upon SaKKH-CBE editing. Pie-chart shows clones with homozygous

C > T (green) mutations and clones harboring a C > A (red), C > G (blue), or indel (orange) on at least one of their alleles. **f** Brightfield images of organoids upon Cas9-homolog multiplexing using SpCas9-ABE to target CTNNB1 and SaKKHcas9-CBE to target *TP53*. Organoids are selected for TP53 mutations by addition of Nutlin-3 to the culture media, and selected for *CTNNB1* mutations by addition removal of CHIR and Rspondin-1. Scale bars are 2000 μm. **g** Brightfield images of Cas9-homolog multiplexing. Organoids are transfected with SaKKH-CBE targeting *TP53* and *APC* and SpCas9-NG targeting *PIK3CA^H1047R^*. Organoids are selected on medium lacking wnt and Rspondin-1 (rpso) and with the addition of Nutlin-3(Nut-3)Scale bars are 2000 μm. **h** Sanger sequencing of the *PIK3CA^H1047R^* locus showing heterozygous mutation induction. PAM is shown in blue and mutation is highlighted with an asterisk. **i** Quantification of editing efficiency of the SpCas9-NG *PIK3CA^H1047R^* sgRNA by Sanger sequencing. Hom homozygous, Het heterozygous, WT wild type. Source data are provided as a Source data file.

multiplexed by just using CBE. To circumvent this issue and expand the flexibility of base editor multiplexing, we explored Cas9 homologs that target broader PAMs and use different sgRNA backbones. The SpCas9 homolog derived from *Staphylococcus aureus* (SaCas9) has been extensively characterized as a highly efficient and significantly smaller alternative to SpCas9. It recognizes an NNGRRT PAM (where R = A or G)[20]. SaKKHcas9 is an evolved variant of SaCas9 and recognizes the more relaxed NNNRRT PAM[18]. We posited that multiplexing SaKKH-CBE together with SpCas9-ABE should allow for simultaneous A > G and C > T edits without sgRNA interference at either target site (Fig. 5a).

First, we designed 6 sgRNAs to introduce premature stop codons in the TP53 coding sequence. We transfected these sgRNAs into hepatocyte organoids together with pCMV-AncSaKKH-BE4Max and functionally selected *TP53* mutants using selection medium containing Nutlin-3. We did not observe any organoid survival after 14 day nutlin-3 treatment in organoids transfected with pCMV-AncSaKKH-BE4Max in combination with sgRNA's targeting Q38* or Q52* in *TP53* while the sgRNA targeting Q317* resulted in some but limited organoid growth (Fig. 5b). Both W146* and Q165* SaKKH-CBE sgRNAs performed reasonably well compared to the previously described Sp-CBE TP53$^{W53*}$ sgRNA (Fig. 5b). Sanger sequencing of 35 Nutlin-resistant clones transferred with SaKKH-CBE W146* sgRNA revealed 27 clones with the correct C > T *TP53$^{W146*}$* mutation (~77%). Interestingly, the remaining clones harbored indel mutations on at least one of their alleles, something we did not observe to this extent with the use of SpCas9-CBE (Fig. 5c, d and Supplementary Fig. 7b, c). Genotyping of 35 Nutlin-resistant clones transfected with pCMV_SaKKH_BE4Max and a *TP53$^{Q165*}$* sgRNA yielded unexpected base editing outcomes of SaCas9. We observed 5 clones harboring C > A and 5 clones harboring C > G mutations on the residue S164 on at least one of their alleles, both introducing stop-codon insertion (Fig. 5e and Supplementary Fig. 8a).

Next, we multiplexed SaKKH-CBE/SpCas9-ABE by co-transfecting both base editors with sgRNAs guiding the SaKKH-CBE towards *TP53$^{W146*}$* and SpCas9-ABE towards the previously described *CTNNB1$^{S45P}$* mutation in hepatocyte organoids. Organoids grew out 14 days after the start of double phenotypic selection by removal of the Wnt activators Chir and R-spondin1, and addition of Nutlin-3 (Fig. 5f). Sanger sequencing for the two loci in selected clones revealed 7 clones out of 12 that carried both TP53 and CTNNB1 mutations (Supplementary Fig. 8b). These results validated our approach for combinatorial gene targeting using Cas9 base editors derived from different species.

Finally, we evaluated the applicability of SaKKH-CBE/SpCas9-ABE multiplexing in colon organoids. In the previous section, we described a hotspot mutation in PIK3CA, E545K, that can be successfully modeled using SpCas9-CBE (Fig. 4d). However, PIK3CA also harbors a second hotspot mutation at the H1047R residue, which is an A > G mutation in exon 21. We aimed to simultaneously inactivate tumor suppressors APC and TP53 and introduce the PIK3CA$^{H1047R}$ mutation in intestinal organoids. We designed 6 SaKKH-CBE sgRNAs targeting APC and co-transfected these with the TP53$^{W146*}$ sgRNA (Fig. 5c) and a SpCas9-NG PIK3CA$^{H1047R}$ sgRNA. We selected all 6 sgRNA combinations for *APC* and *TP53* mutations by removing Wnt-3A and Rspondin-1 from and adding Nutlin-3 to the selection medium. We observed organoid growth after 14 days in all conditions (Supplementary Fig. 8c). We observed 56 clones targeted for APC-sgRNA-3 that carried a stop codon at position 1127 (Q1127*), outperforming the other APC sgRNAs (Fig. 5g and Supplementary Fig. 8c). Next, we clonally expanded 28 double-resistant clones and genotyped the organoids for the PIK3CA$^{H1047R}$ mutation. Sanger sequencing revealed 4 heterozygous and 1 homozygous PIK3CA$^{H1047R}$ mutations out of 28 clones (Fig. 5g, I and Supplementary Fig. 8d). In addition, we observed additional A > G editing on the wobble position of the alanine on position 1046A, resulting in a silent mutation in 10 clones (Fig. 5g, I and Supplementary Fig. 8d). These

results indicate that SaKKH-CBE can be effectively used to introduce stop codons in human ASC-derived organoids. Moreover, while editing outcomes other than C > T can be observed when using SaKKH-CBE, high editing efficiencies can ensure that organoids harboring the desired and targeted mutations can be easily selected using Sanger sequencing.

## One-step generation of a mini-biobank containing mutants that recapitulate complete colorectal cancer tumorigenesis

The previous experiments exemplify the efficiency and versatility of base editing in organoids. Finally, we aimed to model the full complexity of malignant transformation in a single electroporation reaction. We again focused on colorectal cancer which follows a relatively ordered mutational process[55] and can be efficiently modeled through progressive loss of niche factors in ASC-derived colon organoids[13,14]. First, mutations in *APC* render cells Wnt-pathway independent. This is followed by mutational activation of growth pathways, often through mutations in *KRAS* or *PIK3CA*. On several occasions, the TGFβ pathway is affected, e.g., by mutations in *SMAD4*, while *TP53* mutations occur broadly in many cancer types.

We co-transfected wild type colon organoids with a cocktail of sgRNAs, targeting *APC$^{Q1406*}$*, *PIK3CA$^{E545K}$*, *SMAD4$^{R361H}$*, and *TP53$^{W53*}$* with pCMV-AncBE4max. We selected organoids for the first step of tumorigenesis, Wnt-pathway independence. Fourteen days after selection, we observed organoids surviving this selection. Where single *APC$^{Q1406*}$* mutants harbored a strictly cystic morphology, the organoids transfected with the cocktail of 4 sgRNAs showed a wide variety of cystic and dense morphologies (Fig. 6a). We picked 96 clones starting with those with a cystic phenotype and ending at clones with a dense morphology (Supplementary dataset 1). We then genotyped these 96 clones for mutations in the four loci targeted in this experiment. As expected, all 96 clones harbored mutations in *APC*. Two of these contained indels instead of the expected C > T mutation resulting in Q1406* (Fig. 6b, c). Sanger sequencing of *PIK3CA* revealed 66 homozygotes (~69%), 20 heterozygotes (~21%), 2 clones with indels, and the remaining 8 clones were WT (Fig. 6b, c). Sanger sequencing of *TP53* revealed 25 homozygotes (26%), 35 heterozygotes (36%), and the remaining 36 clones were WT (Fig. 6b, c). Lastly, *SMAD4* genotyping showed 12 homozygotes (12.5%), 22 heterozygotes (~23%), one indel, and the remaining 61 clones were WT. Our 96 clonal organoid biobank genetically recapitulated each step of colorectal tumorigenesis as we observed 4 single *APC* mutants, 24 double mutants, 46 triple mutants, and 22 quadruple mutants (Fig. 6d). While *APC/PIK3CA* mutants retained cystic morphology, addition of *SMAD4* and *TP53* mutations caused the organoids to have an increasingly dense and complex phenotype based on bright field images (Fig. 6e).

According to the COSMIC database for somatic mutations in cancer, mutations on the 12th amino acid of the oncogene KRAS are amongst the most common mutations across all cancers. However, due to the lack of a suitable NGG PAM, a conventional SpCas9 cannot be used to introduce this mutation. Thus, we designed a sgRNA that would work with a Cas9 CBE variant that recognizes an NGT PAM (Fig. 6f). To test the efficacy of this sgRNA, we co-transfected the complete set of five sgRNA's into wild-type colon organoids and functionally selected for loss-of-function mutations in *APC* as previously described. We picked and directly lysed 32 clones and genotyped them at the *APC$^{Q1406*}$*, *PIK3CA$^{E545K}$*, *SMAD4$^{R361H}$*, *TP53$^{W53*}$*, and *KRAS$^{G12}$* loci. As expected, all 32 clones harbored homozygous mutations in APC (Fig. 6h). Out of the 32 clones, we observed mutations at the 12th amino acid in 28 clones. Interestingly, we observed two distinct mutations at KRAS G12, namely *KRAS$^{G12N}$* and KRAS$^{G12S}$ (Fig. 6f). Both mutations have been observed in cancer but G12N is much rarer due to the need for two point mutations to induce this SNV. Further genotyping of the other loci revealed a similar

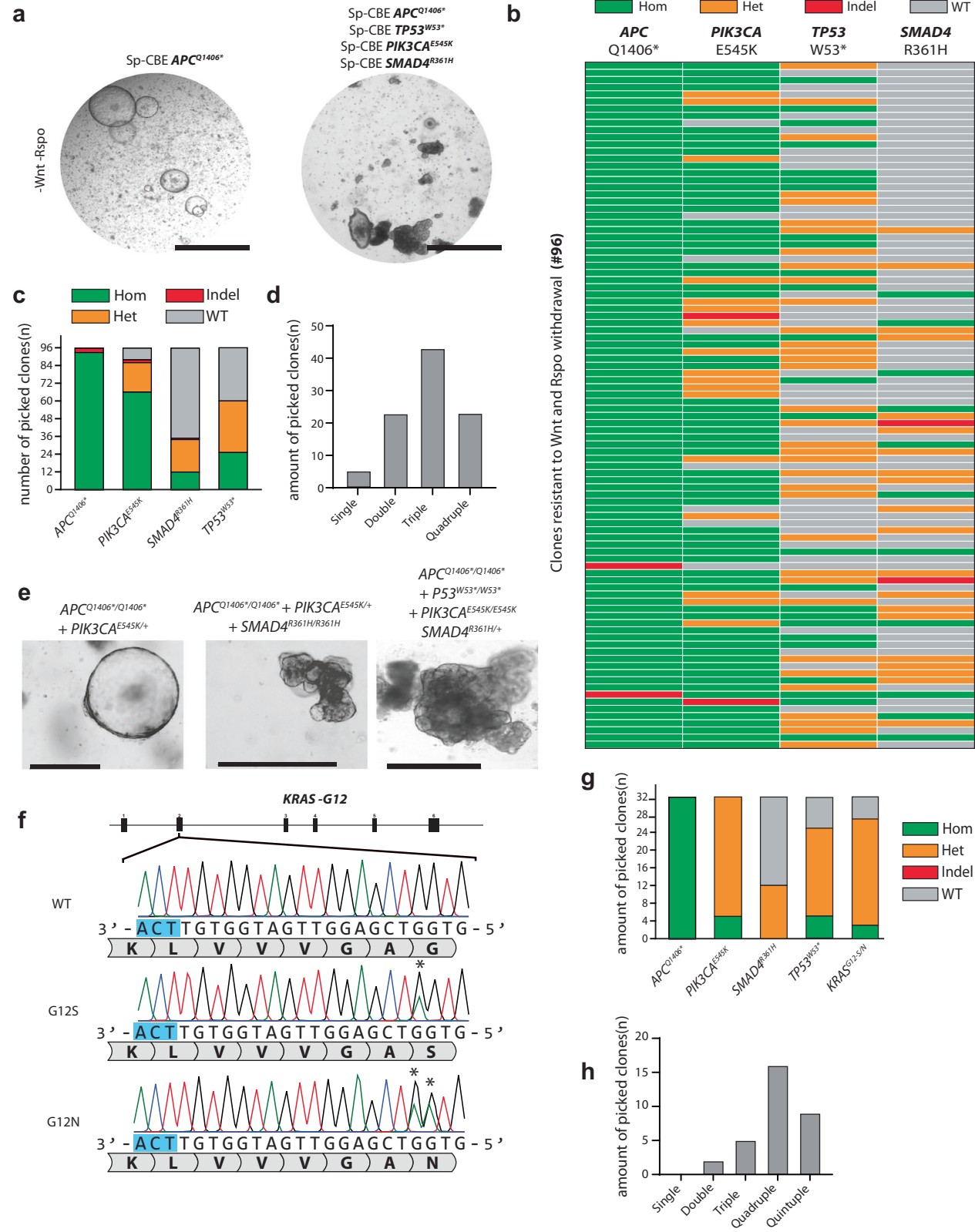

mutational frequency as compared to our previously described quadruple experiment. Out of 32 organoids genotypes, all 32 had a PIK3CA$^{E545K}$ mutation, while 12 had a SMAD4$^{R361H}$ mutation and 24 had a mutation at TP53$^{W53*}$ (Fig. 6g). This data shows that evolved Cas9 variants can be used to increase the target scope of tumor modeling in organoids. By using these Cas9 variants that recognize alternative PAM motifs, organoid models can be created that genotypically reflect the distinct stages of oncogenesis.

## Safety and scope of base editing in tumor modeling

To study potential off-target effects of our multiplexing approach, we performed whole-genome sequencing (WGS) analysis on three colon organoid clones (M1-3) that were co-transfected with SpCas9-CBE and the cocktail of APC/TP53/SMAD/PIK3CA sgRNA's (Supplementary dataset 2). We directly compared these data to sequentially, 2 distinct electroporation events with two different sgRNA's), mutated colon organoids (S1-4).

**Fig. 6 | One-step generation of a mini-biobank containing mutants that reca-pitulate intermittent steps of colorectal cancer tumorigenesis. a** Brightfield images of intestinal organoids transfected with only SpCas9-CBE and a sgRNA targeting *APC^Q1406** and organoids transfected with our cocktail of 4 sgRNA's tar-geting *APC*, *TP53*, *SMAD4*, and *PIK3CA*. Scale bars are 2000 μm. **b** Sanger sequencing of 96 clones surviving Wnt-pathway selection by removal of Rspondin-1 and Wnt-surrogate from culture medium. Starting with a cystic structure (top) to a more dense phenotype (bottom). **c** Quantification of editing efficiency of our 4 sgRNA by Sanger sequencing. **d** Quantification of single, double, triple, and quadruple mutants acquired from a single transfection experiment in human intestinal

organoids. **e** Brightfield images highlighting increasingly dense morphology upon introduction of additional mutations on top of APC. Scale bars are 500 μm. **f** Sanger sequencing traces showing induction of *KRAS^G12S* and *KRAS^G12N* mutations upon using SpCas9-NG in multiplexing five mutations common in colorectal cancer tumorigenesis. The asterisk highlights the edited residue. **g** Quantification of editing efficiency of our 5 sgRNA CBE multiplexing experiment utilizing evolved Cas9 variant SpCas9-NG. **h** Quantification of single, double, triple, quadruple, and quintuple mutants acquired from a single transfection experiment in human intestinal organoids using SpCas9-NG CBE. Hom homozygous, het heterozygous, WT wild type. Source data are provided as a Source data file.

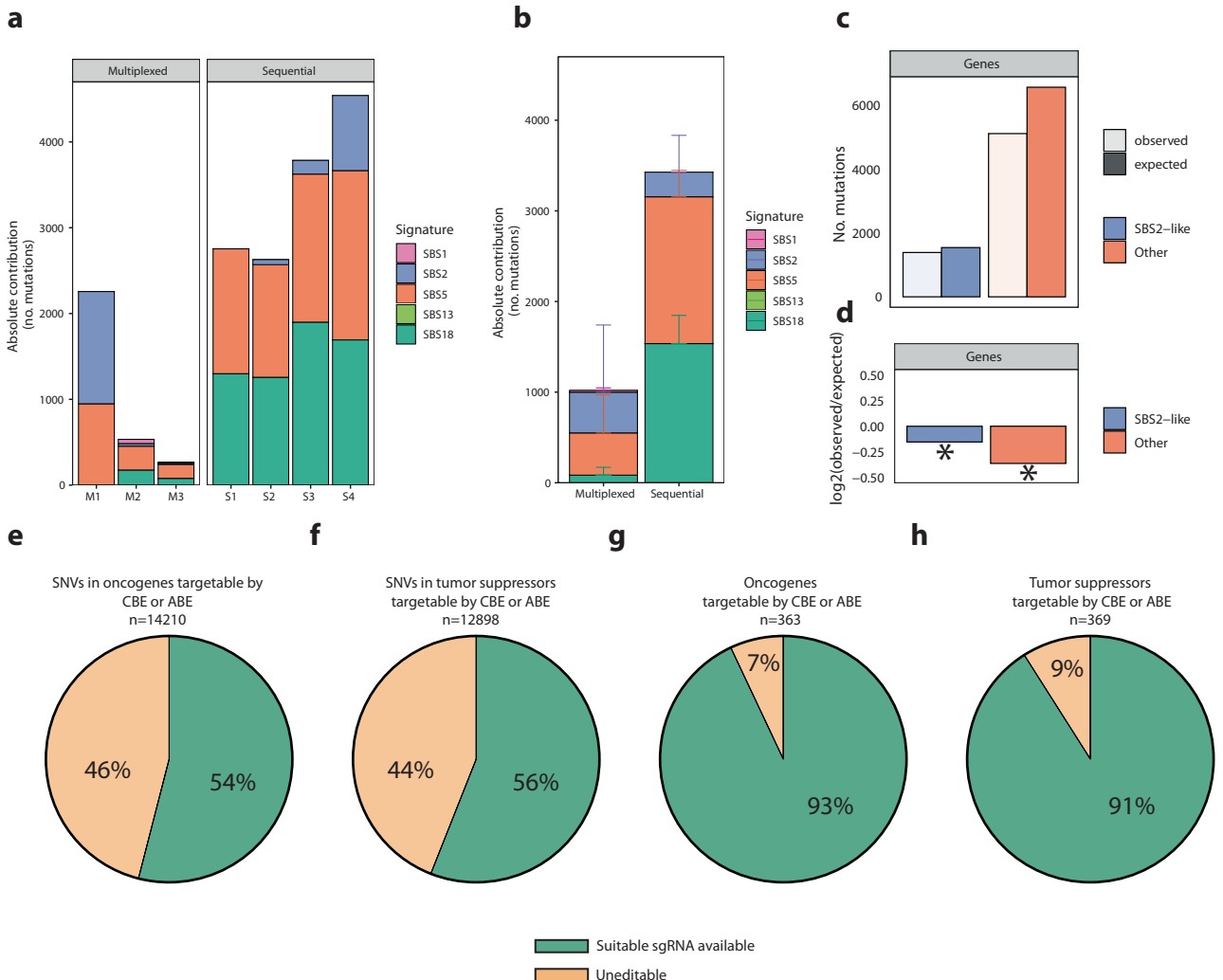

**Fig. 7 | Safety and scope of base editing in tumor modeling. a** Total number of mutations accumulated during either multiplexed or sequential base editor tumor modeling in organoids. Colors in the bars represent mutational signature analysis fit. **b** Average number of mutations in either multiplexed or sequentially edited organoids. Colors in the bars represent mutational signature analysis fit. $n = 3$ independent biological samples in case of the multiplexed experiment and $n = 4$ independent biological samples in case of the sequential experiment. "Data are presented as mean values +/− SD". **c** Number of mutations in gene bodies (light) compared to expected (dark) based on their size and total amount of mutations in

both SBS2-like (blue) and other (orange) observed mutations. **d** log2 normalized depletion scores for both SBS2-like and observed mutations that do not correlate with SBS2. Target scope of adenine and cytosine base editors showing the total amount of pathogenic variants in oncogenes (**e**) and tumor suppressors (**f**). Pie-charts showing the percentage of oncogenes (**g**) and tumor suppressors (**h**) in which at least one pathogenic mutation can be modeled by adenine and cytosine base editors. Asterisks show significance based on binominal testing. Source data are provided as a Source data file.

First, we compared the absolute number of in vitro accumulated mutations in the sequenced clones. We observed a 3.4-fold increase in somatic mutation load in the sequentially (on average 3716) mutated tumoroids compared to the multiplexed clones (on average 1114, Mann–Whitney U one-sided $p = 0.02857$) (Fig. 7a, b). To take a closer look at the cause of these mutations we performed mutational

signature analysis by refitting known in vitro culture- and in vivo APOBEC-related mutational signatures to the mutational profile of the intestinal organoid clones[56–58]. Contribution of four mutational signatures was observed in our clones, of which SBS1 and SBS5, both clock-like signatures associated with age, and SBS18, which is asso-ciated with Reactive Oxygen Species (ROS), are most likely caused by

in vitro culturing of our organoids (Fig. 7a, b)[58,59]. The other signatures, SBS2 and SBS13, has been associated with APOBEC activity in cancer[56] and can be explained by the APOBEC1A bound to Cas9 in the CBE architecture. In addition, when only taking into account the SNVs accumulated due to in vitro culturing (SBS1, SBS5, and SBS18) there is a 5.5-fold ($p = 0.02857$) increased mutational load in sequential versus multiplexed clones. We further analyzed the localization of the observed SNPs. Mutations were less frequently observed in gene bodies than expected based on their size and the total number of mutations (Fig. 7c). However, APOBEC-associated SBS2 was less depleted ($p = 6.368784e−05$) in gene bodies compared to all other mutations ($p = 1.538367e−77$) observed in the sequenced clones (Fisher's exact $p = 0.0005174$) (Fig. 7c, d). This difference may suggest indicate sgRNA-independent off-target effects in single-stranded transcribed genes. To analyze the mutational spread across the genome we created individual rainfall plots (Supplementary Fig. 9). We did not observe any mutational hotspots ('kataegis') that may be caused by off-target sgRNA binding that was not predicted in silico, as would be indicated by a cluster of C > T mutations at lower genomic distances. We performed cancer driver analysis on the sgRNA-independent off-target effects and did not observe any SNVs that was predicted to have high impact on protein function, nor have been described previously as known driver mutations in tumorigenesis (Supplementary Table 4). This indicates that the main differences in mutational load between the two strategies are due to clonal expansion and culturing in vitro rather than due to base editing.

Subsequently, we looked at sgRNA-dependent off-target effects. We did not observe any indels or structural variants in the in silico predicted off-target sites (up to 4 mismatches), or their respective 200 base pair flanking regions. Out of 17,416 SNVs observed in all samples, we observed four mutations that overlapped with the predicted off-target regions. Clone M2 acquired three mutations, two in an off-target spacer sequence and one in a flanking region (Supplementary Table 5). In addition, clone M1 acquired one mutation, just outside of the targeted spacer region. No sgRNA-dependent off-target effects were observed in the sequentially edited cells.

To analyze the impact of this multiplexing approach we took in this study may have on tumor modeling we looked at all targetable mutations (i.e., have a suitable PAM) by SpCas9, SpCas9-NG, and SaKKHCas9 in combination with CBE and ABE. We utilized the COSMIC classification of oncogenes and tumor suppressors and analyzed whether the mutations that are classified as pathogenic would have a suitable sgRNA for base editing[60,61]. Out of all mutations that have been classified pathogenic in oncogenes 7675 (54%) and 7223 (56%) of pathogenic mutations in tumor suppressors can be targeted by either CBE or ABE (Fig. 7e, f). Moreover, one or more pathogenic mutation can be made in 338 (93%) of all described oncogenes and in 336 (91%) of all described tumor suppressors (Fig. 7g, h). Thus, CBE and ABE multiplexing may have a great impact on functional characterization of individual SNVs and the impact of oncogene and tumor suppressor mutation during tumorigenesis. Taken together, these data indicate that while CBEs sometimes result in off-target effects, they remain a safe strategy to create tumor models in organoids. Moreover, by multiplexing this technology, fewer mutations are introduced in the clonal organoid lines compared to when the mutations are introduced sequentially.

## Discussion

In this study, we describe the use of base editors for cancer modeling in human ASC-derived organoids. First, we show that both CBEs and ABEs allow modeling hotspot mutations in the most commonly mutated gene in hepatocellular carcinoma, CTNNB1, and that evolved variants of SpCas9 can be used to increase the targeting space of base editors in organoids. Next, we apply CBE mediated CRISPR-stop for targeted stop-codon introduction in EC-relevant genes in endometrial

organoids. EC remains poorly understood and lacks relevant human models to study early tumorigenesis. By combining different CBEs, we generate a human organoid model of endometrial tumorigenesis. We simultaneously introduce oncogene activation and tumor suppression inactivation mutations using Cas9 homologs and increase the opportunities to generate tumor models even further. We show that with base editor multiplexing, we can generate genetic organoid models for colorectal tumorigenesis in a single step. Lastly, we analyze the off-target effects of base editors and show that significantly less mutations are acquired while multiplexing compared to sequential mutation induction. While base editors have been applied before in organoids to model[36] and repair[31] diseases, we significantly extend their application by demonstrating the opportunities that reside in multiplexing CBE's and ABE's, as well as Cas9 homologs and evolved Cas9 variants. By performing these experiments in hepatocyte, endometrium, and colon organoids, we show the versatility of this genome engineering strategy for generation of 3D models of human tumorigenesis.

We thus demonstrate the versatility and efficiency of base editor multiplexing and envision extrapolation of this strategy towards human ASC-derived organoids of—potentially—all epithelial tissues. This would allow for efficient and quick, one-step generation of libraries/biobanks of organoids harboring combinations of mutations observed in specific tumor types. Alternatively, generation of a library of sgRNAs that target a wide set of oncogene activating and tumor suppressor inactivating mutations would allow for determination of the minimal set of mutations required for tissue-specific transformation. Multiplexed gene-knockout by conventional CRISPR/Cas9-mediated NHEJ is very efficient and can be readily applied in organoids[13,14,62]. However, introduction of oncogene activating mutations depends on the inefficient HDR pathway and remains a challenge[13,14]. Since base editor multiplexing can be done in a single reaction, it outperforms the use of conventional CRISPR/Cas9-mediated genome engineering for tumor modeling.

As a current limitation, CBEs and ABEs only allow for the induction of transition C > T and A > G mutations. Yet, these mutations occur significantly more often in human disease than transversion mutations (62% transition vs 38% transversion[29,63]). However, it should be noted that important tumor driving transversion mutations also occur, e.g., the T > A mutation that causes BRAF(V600E). Development of additional base editors that enable these transversions would be key to unlocking the full potential of base editor multiplexing. Indeed, recent developments have resulted in C > G base editors that use cytidine deaminases in combination with additional fusion proteins that drive repair of the induced U in the DNA towards G instead of T[64]. Of note, in the current study, we occasionally observe these alternative base editing outcomes with SaKKH-CBE, which is likely caused by accessibility issues of UGI to the uracil in the R-loop generated by SaKKH Cas9[64]. Editing outcomes of these C > G base editors are still relatively heterogeneous, but they represent a potentially valuable addition to the base editor multiplexing toolbox.

Even more flexibility for genome engineering without DSB's can be achieved by using prime editing[65]. We have previously used prime editing in hepatocyte and colon organoids to model tumorigenesis[66]. Contrary to base editing, efficient use of prime editing uses two sgRNAs that require significant optimization for efficient use. This results in the need to test multiple sgRNA combinations to find the most effective combination[65,66]. Recent advances in prime editing guide RNA (pegRNA) structure, strategy of pegRNA use, and the development of novel prime editors might result in more generalized sgRNA design which can be extrapolated toward extremely flexible prime editing multiplexing in ASC-derived organoids in the future[67,68].

Safety of genome engineering strategies is always a concern. Previously it has been shown that cytidine but not adenine base editors result in significant, genome-wide, off-target effects[57,69]. By performing WGS on edited clonal organoids we indeed observed genome-wide

APOBEC activity, as indicated by SNVs associated with mutational signature SBS2. While these off-target effects are a downside of this strategy, we argue that a few point mutations are less detrimental to the cell compared to the large-scale chromosomal rearrangements that have been described in the use of conventional Cas9 proteins to achieve desired mutations[27]. In addition, most mutations that we observed were actually caused by processes that are intrinsic to in vitro cell culturing. We show that base editor multiplexing results in less off-target mutations by decreasing the time these organoid clones are kept in culture and the clonal steps required to achieve the desired genotype.

Together, base editor multiplexing in ASC-derived organoids is a versatile tool that allows for generation of tumor models from a wide variety of tissues. As these engineered organoids can be clonally expanded, they could lead to a better understanding of tumorigenesis and potential strategies to develop therapeutic regimens.

# Methods

### Ethics oversight
The study was approved by the UMC Utrecht (Utrecht, The Netherlands) ethical committee, the UMC Leiden (Leiden, The Netherlands) ethical committee, and the Diakonessenhuis ethical committee (Utrecht, The Netherlands) and was in accordance with the Declaration of Helsinki and according to Dutch law. This study is compliant with all relevant ethical regulations regarding research involving human participants

### Organoid culture
Human endometrial organoids were established from tissue biopsies as described elsewhere[32]. Endometrial biopsies were obtained under informed consent from the participants of the study "HUB-Ovarian 14-472**" which was approved by an ethical committee. Established organoids were cultured in a medium containing; Advanced DMEM/F12 (Gibco), 1× Glutamax, 10 mmol/l Hepes, 100 µU/ml penicillin–streptomycin and 1× B27 (All supplied by Thermo Fisher Scientific) plus 15% RSPO1 conditioned medium (home-made), 50 ng/ml EGF (Peprotech), 1.25 mM N-acetylcysteine (Sigma), 0.4 nM Wnt surrogate-Fc Fusion protein[70], 1% Noggin conditioned medium (U-Protein express) 0.5 µM A83-01, and 1 µM PGE2 (Tocris), 100 ng/ml FGF10 (Peprotech) and was supplemented with 100 µg/ml Primocin (Invivogen). expansion medium was refreshed every 2 days. Outgrowing organoids were mechanically passaged with a glass Pasteur pipet every week.

Intestinal organoids are cultured as previously described[2]. In short, the wild-type human colon organoid line P26n, as previously described in Van De Wetering et al. (2015) was cultured in domes of Cultrex Pathclear Reduced Growth Factor Basement Membrane Extract (BME) (3533-001; Amsbio). Domes were covered by medium containing Advanced DMEM/F12 (Gibco), 1× Glutamax, 10 mmol/l Hepes, 100 µU/ml penicillin–streptomycin and 1× B27 (All supplied by Thermo Fisher Scientific), 1.25 mM N-acetylcysteine, 10 µM nicotinamide, 10 µM p38 inhibitor SB202190 (supplied by Sigma-Aldrich). This medium was supplemented with the following growth factors: 0.4 nM Wnt surrogate-Fc Fusion protein[70], 2% Noggin conditioned medium (U-Protein express), 20% Rspo1 conditioned medium (in-house), 50 ng ml EGF (Peprotech), 0.5 µM A83-01, and 1 µM PGE2 (Tocris) and was supplemented with 100 µg/ml Primocin (Invivogen). All organoids were passaged and split once a week 1:6 and filtered through a 40-µm cell strainer (Thermo Fisher Scientific) to remove differentiated structures from the culture. Fetal hepatocyte organoids were cultured as previously described in ref. 33. In short, Domes of cultrex Pathclear Reduced Growth Factor Basement Membrane Extract (BME) were covered with the following expansion medium: Advanced DMEM/F12 (Gibco), 1× Glutamax, 10 mmol/l Hepes, 100 µU/ml penicillin–streptomycin and 1× B27 minus vitamin A (all supplied by

Thermo Fisher Scientific) plus 15% RSPO1 conditioned medium (home-made), 50 ng/ml EGF (Peprotech), 1.25 mM N-acetylcysteine (Sigma), 10 nM gastrin (Sigma), 3 µm CHIR99021 (Sigma), 50 ng/ml HGF (Peprotech), 100 ng/ml FGF7 (Peprotech), 100 ng/ml FGF10 (Peprotech), 2 µM A83-01 (Tocris), 10 mM Nicotinamide (Sigma), 10 µM Rho Inhibitor γ-27632 (Calbiochem), and 20 ng/ml TGFa and was supplemented with 100 µg/ml Primocin (Invivogen). Expansion medium was refreshed every 2 days and organoids were split 1:4 every week.

### Plasmid construction
Human codon-optimized base editing constructs were a kind gift from David Liu; pCMV_ABEmax_P2A_GFP (plasmid #112101; Addgene), pCMV_AncBE4max_P2A_GFP (plasmid#112100; Addgene). pCMV_SpCas9-NG_ABEmax_P2A_GFP, pCMV_SpCas9-NG_AncBE4max_P2A_GFP and pCMV_SaKKH_AncBE4max_P2A_GFP were constructed by PCR amplification (Q5, NEB) amplifying everything except for SpCas9 using pCMV_ABEmax_P2A_GFP and pCMV_AncBE4max_P2A_GFP. Coding sequences for SpCas9-NG and SaKKH were PCR amplified using the following plasmids NG-ABEmax (plasmid #124163; Addgene) and SaKKH-ABEmax (Plasmid #119815; Addgene) that were a kind gift from David Liu. Coding sequences and plasmid backbones were combined using the NEBbuilder HiFi DNA assembly mastermix (NEB) and subsequently transformed using OneShot Mach1t1 (Thermo Fisher Scientific) cells and plasmid identity was checked by Sanger sequencing (Macrogen). The empty sgRNA plasmid backbone for SpCas9 and its derivatives was a kind gift from Keith Joung (BPK1520, Addgene plasmid #65777). Spacer sequences targeting all genes in this study were cloned in the sgRNA plasmid backbone using inverse PCR (Q5 NEB) and subsequently transformed using OneShot Mach1t1 (Thermo Fisher Scientific) cells and plasmid identity was checked by Sanger sequencing (Macrogen). Primer sequences for sgRNA generation can be found in Supplementary Table 3.

### Organoid electroporation and selection
The electroporation protocol for human organoids was adapted from Fujii et al.[71]. In brief, the organoids were maintained in regular culture medium and 2 days prior to the electroporation, the medium was refreshed with the addition of the Rock inhibitor Y-27632. Twenty-four hours before the electroporation, 1.25% (v/v) DMSO was added to the culture medium. On the day of the electroporation, the organoids were recovered from BME with ice-cold medium and enzymatically dissociated to single cells in TrypLE, supplemented with Y-27632, by 2 incubations of 3 min each. After TrypLE inactivation, the single cells mixture (consisting of 106 cells) was washed in Optimem and resuspended in 80 µl of BTXpress supplemented by 8 µg of base editor plasmid (SpCas9-CBE, SpCas9-NG, or SpRY-CBE), 3 µg of sgRNA plasmid, 5 µg of piggyBac carrying hygromycin B resistance plasmid and 5 µg of piggyBac carrying the transposase. In multiplexing experiments, sgRNA concentrations were reduced to 2 µg per sgRNA. The electroporation was performed with the NEPA21 system, with settings described before[71]. After electroporation, cells were resuspended in BME and seeded in a pre-warmed 48-wells plate at 20 µl drop per well. After polymerization, pre-warmed culture medium (supplemented with Y-27632) was added, and the plate was imaged at 24 h to monitor GFP expression. One week after electroporation, selection based on transfection (Hygromycin B gold InvivoGen) or by phenotype was started. APC mutations were selected for by removal of Wnt-Surrogate and Rspo1-conditioned medium, TP53 mutations were selected for by addition of Nutlin-3 and PIK3CA mutations were selected for by addition of Meki to the culture medium.

### Establishment of clonal lines and organoids genotyping
Hygromycin B resistant organoids were manually picked under a microscope and reduced to single cells after which individual clonal lines were established. Few days after splitting, regrowing organoids

from individual clonal lines were picked and subjected to DNA extraction and Sanger sequencing. In brief, the DNA was extracted with the Quick-DNA microprep kit (Zymo Research) following the manufacturer's instructions and used for PCR amplification of the targeted sequence (500–600 bp length) with the Q5 High-fidelity PCR kit (Bio-Labs). PCR products were finally sent for Sanger sequencing (Macrogen) with the forward, reverse or sequencing primer. Genotyping primers can be found in Supplementary Table 4. In silico sanger peak quantification was performed by using Indigo[38].

## Gene expression analysis by QPCR

Organoid RNA was reverse-transcribed and subjected to SYBR Green-based quantitative real-time PCR (qPCR) using the forward and reverse primers described in Supplementary Table 5. Glyceraldehyde-3-phosphate dehydrogenase (GAPDH) and hypoxanthine phosphoribosyltransferase (Hprt) were used as housekeeping genes. Relative gene expression levels were calculated as ΔCt values (Ct 'target' minus Ct 'housekeeping gene') and in most analyses compared between 'sample' and 'reference' to express fold change, i.e., 2-(ΔCt sample − ΔCt reference).

## RNA-sequencing

Mutant organoids were cultured in medium without Wnt/Rspondin for 24 h, following which RNA was extracted using commercially available kit (Qiagen RNeasy). The quality of RNA was checked on an RNA pico chip and only high-quality RNA was used for library preparation and sequencing, which was performed by Macrogen. The raw data was aligned to the human genome assembly using STAR aligner (ref) following read trimming using Cutadapt (ref). RNA-seq analysis was performed using DESeq2 (ref) in R programming language. Genes that were upregulated more than 2-fold were considered differentially expressed. All heatmaps and plots were also made in R.

## Drug sensitivity testing on endometrial organoids

Drug sensitivity assays were performed as previously described[72]. In brief, the organoids were recovered from BME and enzymatically dissociated in TrypLE for 3 min at 37 °C. After inactivation, dissociated organoids were resuspended in BME and allowed to recover for 2 days in a medium containing a low concentration of Egf (0.5 ng/ml) and Fgf10 (1 ng/ml). On the day of dispense, 40 μl of pre-warmed dispase were added per ml of culture medium and the organoids were incubated for 1 h at 37 °C. Next, the organoids were collected and washed with ice-cold Advanced DMEM+++, counted and resuspended at a density of 1000 organoids per well in culture medium containing 5% (v/v) BME and without Egf and Fgf10. We finally dispensed 40 μl per well of the organoids containing medium using a ThermoFisher multidrop dispenser in a 384-wells plate and the drugs were added with an HP D300e digital dispenser. After 5 days of treatment, 30 μl of Cell Titer Glo (Promega) were added to each well and the plate was incubated for 20 min at RT before the luminescence was read with a Tecan SPARK luminescence detector.

## Immunofluorescence and immunohistochemistry

Hepatocyte organoids were harvested using Cell Recovery solution (Corning, Product No. 354253) for 30 min on ice. The organoids were allowed to settle to the bottom of the tube, after which the supernatant was removed and replaced with Formalin. Organoids were fixed for 1 h on ice, washed twice with PBS containing 0.5% Triton X-100, and blocked in buffer supplemented with 0.5% BSA. The samples were incubated with the primary antibody solution (Rabbit anti-CTNNB1, 1:100, Cat#Sc-7199; RRID: AB_634603) overnight at 4 °C and washed three times the next day in PBS-0.5% Triton. They were then incubated with the secondary antibody solution (Goat anti-rabbit Alexa Fluor 647, 1:250; Phalloidin Alexa fluor 488, 1:200; DAPI, 1:1000) for 2 h at room temperature. Following three washed in PBS-0.5% Triton, the

organoids were mounted in Vectorshield (Vector labs) and imaged using a Leica SP8 confocal microscope (63x water objective). Images were saved as LIF files and imported in FIJI (v 1.53) for downstream processing.

Endometrium organoids were fixed overnight in 4% paraformaldehyde at 4 °C followed by dehydration and paraffin embedding. To prepare organoids for histological stainings, intact BME drops containing organoids were collected from the culture plates and incubated in Cell Recovery Solution (Corning, Cat. 354253) on ice for 30 min, occasionally inverting the tube, to dissolve BME. Organoids were then allowed to settle to the bottom of the tube by free gravitation, supernatant removed and the material fixed in 4% paraformaldehyde at room temperature for 1 h. After fixation, the organoids were washed in PBS, and embedded into paraffin blocks. Sections were cut and hydrated before staining. Sections were subjected to H&E staining or immunohistochemistry with antibodies Rabbit anti-human PTEN (Cell Signaling Technology 9552, 1:200) for PTEN and Mouse anti-human p-mTOR (Santa Cruz sc-293133, 1:50) for mTOR. The images were acquired on Leica DM4000 microscope and processed using Leica LAS X software.

## Image analysis

To calculate meal relative intensities, we calculated the intensity profile along lines drawn through individual cells of different organoids using FIJI (v1.53) (Analyze>Plot Profile). We wrote a custom script in R programming language to identify the membrane boundaries for each cell in an unbiased manner using Phalloidin signal as a reference. Once the boundaries were defined, we calculated the mean intensity within the membrane ("plasma membrane") and between membranes ("intracellular") and normalized them to the maximum intensity observed for each genotype. Finally, to plot intensities for cells of different widths on the same axis, we separately normalized the length of all membranes and intracellular regions. All plots were made using the package *ggplot* in R. Statistical significance was calculated using ANOVA followed by Tukey's HSD.

## Whole-genome sequencing and mapping

Genomic DNA was isolated of Matrigel/organoid suspension using the QIAamp DNA Micro Kit (Qiagen), according to protocol. Standard Illumina protocols were applied to generate DNA libraries for Illumina sequencing from 200–500 ng of genomic DNA. All samples were sequenced to 15x base coverage (2 × 150 bp, Illumina NovaSeq 6000). The initial processing of the raw sequence reads was performed using the full analysis pipeline available at https://github.com/UMCUGenetics/IAP. In brief, sequence reads were mapped against human reference genome GRCh38 using the Burrows-Wheeler Aligner v0.7.17 mapping tool (Li and Durbin, 2010 https://pubmed.ncbi.nlm.nih.gov/20080505/), with settings 'bwa mem -c 100 -M'. Duplicate reads were marked using Sambamba v0.6.8 and the Genome Analysis Toolkit (GATK) v4.1.3.0 was used for realignment[73].

## Variant calling and filtering

Next, variants were multisample called with the GATK HaplotypeCaller v4.1.3.0 and GATK-Queue v.4.1.3.0, based on default settings and the additional option "EMIT_ALL_CONFIDENT_SITES." Subsequently, GATK VariantFiltration v4.1.3.0 was used to evaluate the quality of the variant positions, with options -snpFilterName SNP_LowQualityDepth -snpFilterExpression "QD < 2.0" -snpFilterName SNP_MappingQuality -snpFilterExpression "MQ < 40.0" -snpFilterName SNP_StrandBias -snpFilterExpression "FS > 60.0" -snpFilterName SNP_HaplotypeScoreHigh -snpFilterExpression "HaplotypeScore > 13.0" -snpFilterName SNP_MQRankSumLow -snpFilterExpression "MQRankSum <−12.5" -snpFilterName SNP_ReadPosRankSumLow -snpFilterExpression "ReadPosRankSum <−8.0" -snpFilterName SNP_HardToValidate -snpFilterExpression "MQ0 >= 4 && ((MQ0/(1.0 * DP)) > 0.1)"

-snpFilterName SNP_LowCoverage -snpFilterExpression "DP < 5" -snpFilterName SNP_VeryLowQual -snpFilterExpression "QUAL < 30" -snpFilterName SNP_LowQual -snpFilterExpression "QUAL > = 30.0 && QUAL < 50.0" -snpFilterName SNP_SOR -snpFilterExpression "SOR > 4.0" -cluster 3 -window 10 -indelType INDEL -indelType MIXED -indelFilterName INDEL_LowQualityDepth -indelFilterExpression "QD < 2.0" -indelFilterName INDEL_StrandBias -indelFilterExpression "FS > 200.0" -indelFilterName INDEL_ReadPosRankSumLow -indelFilterExpression "ReadPosRankSum <−20.0" -indelFilterName INDEL_HardToValidate -indelFilterExpression "MQ0 > = 4 && ((MQ0/(1.0 * DP)) > 0.1)" -indelFilterName INDEL_LowCoverage -indelFilterExpression "DP < 5" -indelFilterName INDEL_VeryLowQual -indelFilterExpression "QUAL < 30.0" -indelFilterName INDEL_LowQual -indelFilterExpression "QUAL > = 30.0 && QUAL < 50.0" -indelFilterName INDEL_SOR -indelFilterExpression "SOR > 10.0."

Low-quality and subclonal mutations accumulated during clonal expansion of the organoid lines were excluded by annotating using SMuRF release 2.1.5 as described previously[58], (https://github.com/ToolsVanBox/SMuRF). We included all variants in each clone at autosomal or X chromosomes, not present in the bulk control sample that passed VariantFiltration, with a GATK phred-scaled quality score ≥60; minimum base coverage of 5X, a mapping quality ≥30, and a variant allele frequency of at least 0.15[58,74]. Structural variation calling was performed with the GRIDSS-purple-linx pipeline v1.3.2, using the bulk reference sample as 'normal' and a single clone as 'tumor' in each tumor-normal pair[75].

### In silico off target prediction
Potential sgRNA-specific off-target events were predicted using the Cas-OFFinder open recourse tool[76]. All potential off-targets up to 4 mismatches were taken into account, selecting an NGG PAM. Both the potential off-target protospacer regions as well as the flanking 200 bases were considered as regions of interest. Using BEDtools v2.27.1, all variants that passed filtering by SMuRF were intersected with the regions of interest[77]. To control for removal of true variants in clusters, all variants that failed the SnpCluster filter were retrieved as well. In addition, the same potential off-target genomic regions were intersected with all start and end coordinates of the structural variations called by GRIDSS-purple-linx.

### Mutational signature analysis
To assess the genome-wide off-target effects of base editor activity in intestinal organoids, we compared culture-related (SBS1, SBS5, SBS18), as well as APOBEC-related (SBS2, SBS13) mutational signatures to the overall mutational profile using the Bioconductor R package MutationalPatterns v.3.2.0[56,58,59].

### Driver analysis
To exclude the formation of driver mutations by off-target base editor activity, all clone-specific mutations were filtered for predicted effect levels 'HIGH' or 'MODERATE', based on the variant annotation as described above. Subsequently, variants were filtered for presence in the Cancer Gene Census (COSMIC, v2019-05-09).

### Quantification and statistics
No statistical methods were used to predetermine sample size. The experiments were not randomized and the investigators were not blinded to the sample allocation during experiments and outcome assessment. Value of n is always displayed in the figure as individual data points, and in the legends. Sanger sequencing validation for editing efficiency was always performed in clones derived from at least two individual transfection experiments to ensure reproducibility. Statistical analysis was performed with the GraphPad Prism 9 software. And R (v.4.1.3).

### Figure schematics
All schematics in this manuscript are created using Adobe Illustrator 2023. Part of Fig. 1c was created with the use of Biorender.com.

### Reporting summary
Further information on research design is available in the Nature Portfolio Reporting Summary linked to this article.

## Data availability
The whole-genome sequencing data from this publication have been deposited to the European Genome-phenome Archive under accession code: EGAS00001006886 under restricted access according to our ethics regulations. Access to the whole genome sequencing data is available upon reasonable request to either the corresponding authors or the Data Access Committee of the biobank of the Prinses Maxima Centrum in Utrecht, The Netherlands. The corresponding authors typically start the data access process within 5 working days. After access has been granted, data can be downloaded for further analysis. The RNA sequencing data from this publication have been deposited to the Gene Expression Omnibus, under accession code: GSE236490. All other data are available in the main text, its Supplementary Information files or in the source data file, or from the corresponding authors upon reasonable request. Source data are provided with this paper.

## Code availability
All software tools used for sequencing data analysis can be found online at: https://github.com/ToolsVanBox. The code that we have used to calculate the targeting scope of base editors in tumor modeling can be found in Supplementary code 1.

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

## Acknowledgements

We thank Stieneke van den Brink for the production of R-spondin1-conditioned medium, essential for the organoid expansion medium. This work was supported by an award from the Cancer Research UK Grand Challenge (C6307/A29058)) and the Mark Foundation for Cancer Research to the SPECIFICANCER team (H.C., M.G.) and by the Netherlands Organ-on-Chip Initiative, an NWO Gravitation project (024.003.001) funded by the Ministry of Education, Culture and Science of the government of the Netherlands. M.B. is a postdoctoral researcher supported by a long-term EMBO fellowship (ALTF 769-2019), S.G. was supported by the Schmidt Science Fellowship. Part of Fig. 1c was created with the use of Biorender.com.

## Author contributions

M.H.G., S.G., and M.G.B. designed the study, performed experiments, and wrote the manuscript. N.A., M.C., and P.S. performed sanger-based genotyping reactions and organoid cell culturing. L.L.M. and F.P. were responsible for WGS-based off-target analysis under the supervision of R.v.B. G.J.F.S. and A.A.R. performed RNA sequencing. S.H.S. and D.H. aided in experimental design and data analysis regarding hepatocyte and endometrium experiments. H.B. performed immunohistochemistry. S.M.C.d.S.L. aided in securing material for hepatocyte organoids. This study was performed under the supervision of J.H.v.E. and H.C.

## Competing interests

H.C. is an inventor on several patents related to organoid technology; his full disclosure is given at https://www.uu.nl/staff/JCClevers/. H.C. is currently head of pharma Research Early Development (pRED) at Roche. H.C. holds several patents on organoid technology. Their application numbers, followed by their publication numbers (if applicable), are as follows: PCT/NL2008/050543, WO2009/022907; PCT/NL2010/000017, WO2010/090513; PCT/IB2011/002167, WO2012/014076; PCT/IB2012/052950, WO2012/168930; PCT/EP2015/060815, WO2015/173425; PCT/EP2015/077990, WO2016/083613; PCT/EP2015/077988, WO2016/083612; PCT/EP2017/054797, WO2017/149025; PCT/EP2017/065101, WO2017/220586; PCT/EP2018/086716, n/a; and GB1819224.5, n/a. The remaining authors declare no competing interests.
