## [Peer Review File · Nature Communications]

One-step generation of tumor models by base editor multiplexing in adult stem cell-derived organoidsREVIEWER COMMENTS

Reviewer #1 (Remarks to the Author):

Overall Summary of the findings

Geurts et al. applied a recently evolving base-editing tool to explore A>G, C>T mutations associated with oncogenesis using adult stem cell-derived organoid systems. For the most part, individual gene targeting was validated using conventional and evolved variants of Cas9-CBEs and -ABEs in hepatocyte, endometrial and intestinal organoid systems. Some of the mutagenic (either activating or deactivating) properties were further confirmed by biological experiments such as niche factor deprivation and pharmacological experiments. Finally, by multiplexing SaKKH-CBE together with SpCas9-ABE, the authors successfully introduce oncogenic activating/tumor suppressor inactivating mutations simultaneously in intestinal organoids to make a genetically diverse mutagenic organoid panel. This experimental toolset moves tumoroid investigation to the next level.

Major Concern

- Variable targeting efficiency: There are appreciable amounts of off-target effect or variable efficiency. This can be simply recognized by Figure7b showing much less efficiency in TP53 and SMAD4 editing. How these variations are derived from? These can be caused by several parameters, including the consensus sequence preference/exposure time of the deaminase, and the binding efficiency of the sgRNA to the protospacer. It would be informative to explain where such variations come from, preferably with some experimental supporting data.
- Insufficient biological validation: For example, Fig4j-k and Supplementary Figure2g-j are extremely unclear as to how they determine viability and organoid survivability AUC. It might make more sense to determine IC 50 or show dose-response curve to support the conclusion. Similarly, Fig6 hepatocyte organoid experiments are only supported by low-resolution pictures and qualitative description of the acquired phenotype without relation to homo/hetero mutational carriage. Similarly, in Fig7, no further characterization was made on tumor-like organoids. Since the mutagenized organoid show aberrant morphology, it might be informative to profile how their epithelial identity is disrupted and how

malignancy-related traits are different across single to quadruple clones.

Minor Concern

Grammatical errors and redundant sentences are found at times and it is encouraged for an additional round of proofreading process to improve the readability of the article.

- Fig1 seems published design and doesn't need to stay as a standalone main figure panel. It may confuse the audience as if this base editing complex is novel.
- Page3: There are no descriptions about targeting D32 residue while other residues are explained. Perhaps the last sentence in the previous paragraph covers this point, but if so, please specify for better readability.
- Figure2 b, sgRNA-2 target position is at 124 as opposed to 121. Please correct
- Figure2 f, g call out is not correctly referred in the manuscript.
- In the intro, base editors are used to target within the size of four nucleotides between positions 4 and 8 from the 5' end of the protospacer. However, in Figure2E, SpCas9-NG-ABEmax (CTNNB1 : T41A) generated mutation in position3 from the 5'end and also supplemental figure 4d generates in position9. Please discuss.
- In page 5, Please delete "the" the plasma membrane. Correct "mutantions"
- In page 7, "ACS"-derived organoids. Is it ASC?
- Figure4b x-axis labeling is disorganized.
- Why SpCas9-NG-CBE wouldn't make R130 mutation? Please discuss.
- In page 7, mention around drug screen in page7 is awkward since authors tested only two drugs.
- In page9 and 11, please correct "suppressor"
- Fig5 legend: please correct "withdrawel"
- In page11, call out of figure 5h has wrong clone number (#20 instead of 16)
- In page13, SpCAs9 needs to be corrected to SpCas9
- Figure 5d call out should be 5e?
- Rationale to shift into hepatocytes is unclear in Fig6. Also Fig6c-e needs to show homo/hetero mutant distribution as in the other figures.
- Conflict of interest is not addressed by simple providing the website.

Reviewer #2 (Remarks to the Author):

Geurts et al demonstrated the introduction of disease mutations into human hepatocyte, endometrial and intestinal organoids using CRISPR base editors. They did not develop any new CRISPR base editing technologies but showed rigorous analyses of using the existing base editors in organoid models, which I appreciate. I do not have any major criticisms of their experiments other than what I list below. However, one major concern is that despite that the authors state the usefulness of organoid models with engineered disease mutations in the interrogation of disease phenotype, none of such analyses is performed beyond the validation analysis of expected phenotypes. Other detailed comments follow:

Major comments:

1. Line 245 “On the other hand, organoids that were functionally selected for mutations in APC (removal of Wnt) and TP53 (addition of Nutlin-3) but genotyped for the PIK3CA-E545K mutation revealed 8 homozygous (50%) and 6 heterozygous (37.5%) mutations out of 20 clones for a total mutation induction of 87.5% (figure 5h). These results demonstrate the versatility of CBEs and ABEs in introducing point mutations in both tumor suppressor and oncogenic activator genes.” What was the hypothesis here? Isn't this obvious? Also, “suppressor.”
2. Several base editing papers have been published to induce mutation in organoid cultures already. For example, Dow lab has demonstrated that an optimized cytidine base editor construct efficiently induces C->T changes in intestinal organoids (Zafra et al 2018 Nature Biotechnology). What is the advantage and novelty of the current work compared to the previous works?
3. Fig 2c is a bit confusing. In line 101 “By co-transfecting ABEs or CBEs with sgRNAs and two plasmids that enable stable integration of a hygromycin cassette”: Fig 2c shows four separate plasmids and this sentence is unclear that they used four independent plasmids. In the Materials and Methods section (line 464), it says “8 µg of base editor plasmid (SpCas9-CBE, SpCas9-NG or SpRY-CBE), 3 µg of sgRNA plasmid, 5 µg of piggyBac carrying hygromycin B resistance plasmid and 5 µg of piggyBac carrying the transposase”. Why did the authors use four different plasmids? The plasmid having hygromycin B resistance can be independently acquired into a cell. Is there any evidence that the hygro selection here

works as a marker to select the all-in-one cells? I would propose to minimize the number of plasmids. For example, it could be like: [5' ITR]-[constP]-[base editor]-[polyA signal]-[U6p]-[gRNA]-[constP]-[Hygromycin]-[polyA signal]-[3' ITR].

4. Related to the previous point. Did the authors perform any analysis that showed the hygro selection contributed to the enrichment of base-edited cells?
5. Fig 3a-e, are they cherry-picked examples? These panels can be grouped just as “panel a.” Similarly in Fig 3f,g, how many organoids are analyzed here? Are they also arbitrary picked results or population averages of many?
6. Cancer cells are known to be heterogeneous even though they have the same genotype. When the authors analyzed the CTNNB1 cellular localization harboring the mutation alleles, did they observe any clone-to-clone differences in CTNNB1 localization?
7. Fig 1, Fig 4a, and Fig 6a do not show any new knowledge and can be omitted.
8. Fig 3i: Needs statistical tests.
9. In Fig 4d: how were the mixed signals from heterozygous mutations treated here?
10. In Fig 4g,h,j: Need statistical tests.
11. Fig 6b: please show all the gRNA datasets.
12. Fig 6f: Does the one with TP53 mutation have any growth phenotype?

Minor comments

13. Line 71 “As base editors mediate genetic changes without the need for deleterious DSBs”: they often induce deletions.
14. Line 98 “To model hepatocellular tumorigenesis in human hepatocyte organoids, we designed, three sgRNAs to introduce oncogenic mutations in the CTNNB1 locus”: It is better to clarify that the genome-editing reagents were introduced to the organoids but not to ASCs.
15. Line 164 “We first used SpCas9-CBE to target the Q245 residue in exon 7 of PTEN”: Was this also to stem cells or to the organoids?
16. Line 94 “This is followed by a GSK3-mediated phosphorylation cascade that starts at threonine-41 (T41) and the subsequent phosphorylation of serines at position 33 and 37 (S33 and S37)”: Need a reference.
17. Fig 2d-g should be combined into a single panel. Please specify the gRNA spacers and their expected target windows.

18. Fig 3a-e should be combined into a single panel
19. Fig 4a is unnecessary but the codon conversation table is good to keep.
20. Fig 4b: x- and y-axis labels are not aligned
21. Fig 4h-j: Maybe it is better to unify the color codes of the bar charts.
22. Fig 7b and c are the same data.
23. Fig 7d: Please use bar chart instead of line chart.
24. Fig 1a: Although I think this figure does not need to be presented, the PAM is not indicated in the diagram.
25. Line 467-468: Include the voltage information for the electroporation.

Reviewer #3 (Remarks to the Author):

Geurts, Boretto, Shashank Gandhi and colleagues present the application of base editors for the generation of tumor models in human ASC-derived organoids from multiple tissues. They show the efficacy of both cytosine and adenine base editors and use them to model hot-spot point mutations in CTNNB1, PTEN and PIK3CA APC and TP53. Moreover, they model colorectal cancer tumorigenesis by targeting 4 cancer genes in a single experiment. This is an interesting study that reports the utility of base editors as a technology to model mutations in driver genes in ASC-derived organoids and generate biobanks of organoids harboring combinations of mutations observed in specific tumor types. The methods and the conclusions will interest cancer biologists. However, to solidify the value of their study, the authors should address few concerns.

Major

- 1) As stated by the authors, undesired byproducts of base editing, occur at the target site. This happens because the base editors can generate bystander edits when multiple C or A nucleotides are present within the 4-8 nucleotides editing window. In addition to on-site byproducts, undesired deaminations at off-target DNA sites can also occur (in both Cas9-dependent and independent manner). Have the authors performed any assay (for example CIRCLE-seq) for detecting the extent of such off-targets?
- 2) Hepatocyte organoid outgrowth upon Nutlin-3 selection on 3 SaKKH-sgRNAs (W146*, Q165* and Q317*) was compared to a scrambled control sgRNA and to the optimal TP53-

W53* sgRNA in combination with SpCas9-CBE. Is the difference in growth due to the performance of the SpCas9-CBE vs SaKKHCas9-CBE? Since the authors are comparing different nucleotides (that can suffer from chromatin accessibility effect) it's hard to distinguish between the Cas9 performance vs local effect.

3) Limitations of the Study: Please expand the discussion regarding the limitations of using the base editor system as a tool to generate bio banks of ASC-derived organoids. As stated by the authors, this approach doesn't allow to reproduce the 38% of transversion. Which is the percentage of mutations occurring in oncogenes and tumor suppressors that is actually possible to model with this approach?

4) Since the engineered organoids represent an excellent model to a better understand the tumorigenesis and develop now therapeutic regimens, it would be desirable to compare the transcriptome profile of the organoid models for colorectal tumorigenesis with patients-derived organoids with matching mutational profile.

5) Could the authors comment how their results compare to other papers with a similar approach (for example PMID: 29969439).?

Minor

1) Figure 2i. In the text is stated that all the mutant organoids were tested for the expression of Wnt-pathway genes. Since the error bars are not shown, it's not clear the variability between the mutants. Or only one mutant organoid is represented? Please specify

2) Figure 4b. The numbers are not clear

3) Supplementary Figure 2a. Please correct the caption "brightfield" (overlapping with anchor)

4) Supplementary Figure 2d. Specify in figure legend WT, HM, HT. axis missing.

5) Supplementary Figure 2g,h, j. Y axis missing

6) Figure 4h – "Bar graphs showing decreased gene expression of PTEN". The downregulation of PTEN is not clear in the Q130* mutant

7) Supplementary figure 3f: Did the authors mean "TP53 W54* sgRNA" instead of "scramble sgRNA"?

8) What is the overall % of successfully electroporated ACS in the different tissues?

9) Capital letters in figure legends are not consistent. Please, correct.

Reviewer #1 (Remarks to the Author):

Overall Summary of the findings

Geurts et al. applied a recently evolving base-editing tool to explore A>G, C>T mutations associated with oncogenesis using adult stem cell-derived organoid systems. For the most part, individual gene targeting was validated using conventional and evolved variants of Cas9-CBEs and -ABEs in hepatocyte, endometrial and intestinal organoid systems. Some of the mutagenic (either activating or deactivating) properties were further confirmed by biological experiments such as niche factor deprivation and pharmacological experiments. Finally, by multiplexing SaKKH-CBE together with SpCas9-ABE, the authors successfully introduce oncogenic activating/tumor suppressor inactivating mutations simultaneously in intestinal organoids to make a genetically diverse mutagenic organoid panel. This experimental toolset moves tumoroid investigation to the next level.

A: We thank the reviewer for the positive comments on our manuscript. We have gone through all the comments made by the reviewer and have addressed these at different places throughout the manuscript. We have described these changes below.

Major Concern

- Variable targeting efficiency: There are appreciable amounts of off-target effect or variable efficiency. This can be simply recognized by Figure 7b showing much less efficiency in TP53 and SMAD4 editing. How these variations are derived from? These can be caused by several parameters, including the consensus sequence preference/exposure time of the deaminase, and the binding efficiency of the sgRNA to the protospacer. It would be informative to explain where such variations come from, preferably with some experimental supporting data.

A: Well taken. As is known, a variety of parameters can affect guide RNA efficiency, including accessibility of the target site, composition of the protospacer, undocumented SNPs in the genome, etc. On top of that, base editing is only efficient within a "small window" of 4 nucleotides (as extensively discussed by the David Liu lab) with an abundance of supporting data. However, we agree with the reviewer that understanding the potential off-target effects of our tools is important when it comes to tumor modeling. Therefore, we performed whole genome sequencing of several clones obtained from the experiment described in figure 7. This was an appropriate platform to test off-target activity given that several guides were simultaneously electroporated in intestinal organoids. However, we do not find evidence for off-target effects induced specifically by the guide RNAs used in this study, supporting the evidence presented in this study that these tools can effectively model different tumors across several different organoid lines.

- Insufficient biological validation: For example, Fig 4j-k and Supplementary Figure 2g-j are extremely unclear as to how they determine viability and organoid survivability AUC. It might make more sense to determine IC₅₀ or show dose-response curve to support the conclusion. Similarly, Fig 6 hepatocyte organoid experiments are only supported by low-resolution pictures and qualitative description of the acquired phenotype without relation to homo/hetero mutational carriage. Similarly, in Fig 7, no further characterization was made on tumor-like organoids. Since the mutagenized organoid show aberrant morphology, it might be informative to profile how their epithelial identity is disrupted and how malignancy-related traits are different across single to quadruple clones.

A: We have revised this section of the manuscript accordingly. First, we further refined our system by introducing PIK3CA^{E545K} mutation in endometrial organoids (PTEN^{Q245} mutant). PIK3CA mutations are extremely frequent in endometrial cancer, and it is not uncommon to have multiple mutations along the PI3K pathway (PIK3CA and PIK3R1 are mutually exclusive while PTEN and PIK3CA or PIK3R1 are often co-mutated). It is now possible to see how the addition of a second mutation along the pathway strengthens the resistance towards these inhibitors and makes our experimental model more robust.*

We have also removed panel 'K' from figure 4 and replaced the heatmaps with dose-response curves, as suggested by this reviewer. We have also replaced the bar graphs showing the AUC with ones that show the IC₅₀ for the inhibitor in each genetic background. Following suggestions by the reviewers, the revised figures 4g-h now show the dose-response curve of the mTOR inhibition through Everolimus treatment (Fig.

4g) and the bar graph reporting the IC50 of the inhibitor (Fig. 4h). Also, the revised Supplementary Figure 2i-l shows the dose-response curves to AZD-4547 (Supplementary Fig. 2i) and IC50 (Supplementary Fig. 2j), the dose-response curve to Alpelisib (Supplementary Fig. 2k) and IC50 (Supplementary Fig. 2l). We believe that the effect of these mutations on lowering the sensitivity of the organoids to different PI3K pathway inhibitors is thus clearly evident. Finally, we acknowledge the reviewer's comments about figure 7.

Finally, we report a set of genome engineering tools and approaches that allow researchers to generate complex cancer models with ease, thereby pushing the boundaries of human tumor modeling. Characterizing how individual or combination of mutations influence malignancy-related traits is interesting but -we feel- is beyond the scope of this manuscript. However, we recognize the reviewer's opinion and have now added more discussion on the characterization of different mutational backgrounds in organoids.

Minor Concerns

Grammatical errors and redundant sentences are found at times and it is encouraged for an additional round of proofreading process to improve the readability of the article.

A: We have now gone through the manuscript and made several changes that we believe have helped improve the readability of this paper.

- Fig1 seems published design and doesn't need to stay as a standalone main figure panel. It may confuse the audience as if this base editing complex is novel.

A: The purpose of figure 1 was to orient the reader to the different classes of base editors used in this study. We make it very clear that the base editing complex is not novel. However, to allay this reviewer's concerns, we have now moved this figure to the supplementary data. Additionally, we have now added additional text in the figure legend that the base editing complexes have been adapted from the work of David Liu and colleagues.

- Page3: There are no descriptions about targeting D32 residue while other residues are explained. Perhaps the last sentence in the previous paragraph covers this point, but if so, please specify for better readability.

A: We have now added text that directly talks about targeting the D32 residue.

- Figure2 b, sgRNA-2 target position is at 124 as opposed to 121. Please correct.

A: According to our database-research the T41A mutation is caused by c.121A>G (<https://www.ncbi.nlm.nih.gov/clinvar/RCV000019143/>)

- Figure2 f, g call out is not correctly referred in the manuscript.

A: Well taken and corrected.

- In the intro, base editors are used to target within the size of four nucleotides between positions 4 and 8 from the 5' end of the protospacer. However, in Figure2E, SpCas9-NG-ABEmax (CTNNB1 : T41A) generated mutation in position3 from the 5'end and also supplemental figure 4d generates in position9. Please discuss.

A: While the base editing window is indeed between positions 4 and 8 from the 5' end of the protospacer, it is not uncommon to sometimes observe edits just outside this editing window, if a suitable nucleotide is present. We have added a brief discussion about this in the manuscript.

- In page 5, Please delete "the" the plasma membrane. Correct "mutantions"

A: Thanks. We have now fixed these errors.

- In page 7, “ACS”-derived organoids. Is it ASC?

A: Indeed, it was meant to be ASC-derived. We have fixed this error.

- Figure 4b x-axis labeling is disorganized.

A: The axis in figure 4b has been corrected and the percentages of PTEN mutant samples are now visible. Also, since we removed Figure 4a in agreement with various comments, Figure 4b is panel 'A' in the revised manuscript.

- Why SpCas9-NG-CBE wouldn't make R130 mutation? Please discuss.

A: ABEs and CBEs have different efficiencies, as discussed in response to one of this reviewer's previous comments. Amongst these is the C/A editing position within the protospacer sequence and the tri-nucleotide sequence at the editing position, as previously reported (See Kim F. Marquart et al., *Nature Communications* volume 12, Article number: 5114 (2021); Fig. 1c-d, Fig. 1g-h). In particular, higher editing efficiency with both CBEs and ABEs are observed at position 5-6-7 of the protospacer rather than at position 3-4 and 8. In this regard, both sgRNAs which rely on the NGN PAM are targeting a Cytosine in position 3 (sgRNA-1) or 8 (sgRNA-2). As for the tri-nucleotide containing the editing site, the most favorable outcome using a CBE derives from targeting TCA, TCG, TCT triplets. However, we are targeting the ACG triplet, which has been shown to have lower editing efficiency outcomes (See Kim F. Marquart et al., *Nature Communications* volume 12, Article number: 5114 (2021); Fig. 1h). By using sgRNA-3 (combined with the SpRY-CAS9, with NAN PAM) which targets the Cytosine at position 5 of the protospacer sequence, we improve the editing efficiency. Finally, in our hands, base editors that identify an NGG PAM are highly efficient compared to those that identify the NGN PAM. Nevertheless, by simply increasing the number of clones picked and genotyped, we might observe editing with both sgRNA-1 and sgRNA-2. However, it is our opinion that sgRNA-3 represents a better, cost-effective strategy to target this locus.

To better help the reader understand these differences, we now discuss this distinction in the manuscript together with important references that reported the activity of various base editors employed in this study. We hope that this clarifies the point for the reviewer.

- In page 7, mention around drug screen in page 7 is awkward since authors tested only two drugs.

A: In this manuscript, we have tested 8 different concentrations of 3 inhibitors on 3 different genetic backgrounds (WT, PTEN^{Q245*}, and PTEN^{Q245*}/PIK3CA^{E545K}), which we believe constitutes a drug screen. Yet, we have reworded this and now use the term drug sensitivity testing.

- In page 9 and 11, please correct “suppressor”
- Fig 5 legend: please correct “withdrawal”
- In page 11, call out of figure 5h has wrong clone number (#20 instead of 16)
- In page 13, SpCas9 needs to be corrected to SpCas9
- Figure 5d call out should be 5e?

A: These errors have been fixed in the revised manuscript.

- Rationale to shift into hepatocytes is unclear in Fig 6. Also Fig 6c-e needs to show homo/hetero mutant distribution as in the other figures.

A: In our hands, hepatocyte organoids present a reliable platform to quickly test the efficiency of individual guide RNAs given the rate of growth and ease of passaging. Since we had to optimize a different base editor for the experiments we described in figure 6, we decided to test the system in hepatic organoids. In fact, we encourage readers to consider using hepatic organoids for quickly testing the efficacy of their genome engineering tools. We have now clarified this in the text.

A: Figure 6c-e is kept as is in the current version of the manuscript. We have further clarified these pie-charts in the text as well as the legend of the figure by addition of the sentence "on at least one of their alleles".

• Conflict of interest is not addressed by simply providing the website.

We have added a more extensive conflict of interest statement in the current version of the manuscript

Reviewer #2 (Remarks to the Author):

Geurts et al demonstrated the introduction of disease mutations into human hepatocyte, endometrial and intestinal organoids using CRISPR base editors. They did not develop any new CRISPR base editing technologies but showed rigorous analyses of using the existing base editors in organoid models, which I appreciate. I do not have any major criticisms of their experiments other than what I list below. However, one major concern is that despite that the authors state the usefulness of organoid models with engineered disease mutations in the interrogation of disease phenotype, none of such analyses is performed beyond the validation analysis of expected phenotypes. Other detailed comments follow:

A: We acknowledge -of course- that the tools developed in this study are not novel and did not intend to create that impression. However, their application towards efficient tumor modeling in three different tissue backgrounds is the crux of this study, which we believe will allow the scientific community at large to design more elaborate experiments in their labs. Investigating disease phenotypes across all the models generated in this study would go beyond the scope of this manuscript.

Major comments:

1. Line 245 "On the other hand, organoids that were functionally selected for mutations in APC (removal of Wnt) and TP53 (addition of Nutlin-3) but genotyped for the PIK3CA-E545K mutation revealed 8 homozygous (50%) and 6 heterozygous (37.5%) mutations out of 20 clones for a total mutation induction of 87.5% (figure 5h). These results demonstrate the versatility of CBEs and ABEs in introducing point mutations in both tumor suppressor and oncogenic activator genes." What was the hypothesis here? Isn't this obvious? Also, "suppressor."

A: This sentence was awkwardly phrased in the original manuscript. We performed this experiment to assess the efficiency of the PIK3CA E545K sgRNA. This is now rephrased in the current version of the manuscript.

2. Several base editing papers have been published to induce mutation in organoid cultures already. For example, Dow lab has demonstrated that an optimized cytidine base editor construct efficiently induces C->T changes in intestinal organoids (Zafra et al 2018 Nature Biotechnology). What is the advantage and novelty of the current work compared to the previous works?

A: We thank the reviewer for this comment and for pointing out the work of our colleagues from the Dow lab. We do not refute that other labs (including our own lab) have shown the efficiency of base editors in organoids. However, the scale of our study is significantly different. As the reviewer might know, creating cancer models that faithfully recapitulate the genetics of human cancer has been quite challenging, especially when genes harbor specific gain/loss-of-function mutations at multiple loci. Sequential introduction of such tumorigenic mutations is laborious and can sometimes take several months to obtain mutant organoids, a major limitation that was holding the field back. Our work addresses this limitation. By creating complex combinations of the exact changes associated with cancers in a single transfection reaction, we demonstrate the versatility and flexibility of multiplexed CRISPR base-editing for cancer modeling, and we do this across several human epithelial tissues: endometrium, colon, and liver organoids. We envision that our approach can be readily adapted to create in vitro models for tumorigenesis of solid human tumors for multiple other tissues. We have addressed this in the introduction and the discussion in the current version of the manuscript.

3. Fig 2c is a bit confusing. In line 101 "By co-transfecting ABEs or CBEs with sgRNAs and two plasmids that enable stable integration of a hygromycin cassette": Fig 2c shows four separate plasmids and this sentence is unclear that they used four independent plasmids. In the Materials and Methods section

(line 464), it says “8 µg of base editor plasmid (SpCas9-CBE, SpCas9-NG or SpRY-CBE), 3 µg of sgRNA plasmid, 5 µg of piggyBac carrying hygromycin B resistance plasmid and 5 µg of piggyBac carrying the transposase”. Why did the authors use four different plasmids? The plasmid having hygromycin B resistance can be independently acquired into a cell. Is there any evidence that the hygro selection here works as a marker to select the all-in-one cells? I would propose to minimize the number of plasmids. For example, it could be like: [5' ITR]-[constP]-[base editor]-[polyA signal]-[U6p]-[gRNA]-[constP]-[Hygromycin]-[polyA signal]-[3' ITR].

A: The approach that the reviewer recommends is indeed elegant. However, the number of plasmids used in a single experiment does not constrict our ability to positively select for point mutations in either of the three tissues we present in this study. Therefore, we believe that it would be useful to adapt into a single-plasmid strategy, only if an application demands so. Notably, our multiple plasmid-approach is more flexible as this allows us to clone individual sgRNA vectors once and then use them in different combinations.

The hygromycin resistance strategy here is used solely to select for transfected cells. Our data indicates (and others from our lab have shown: <https://pubmed.ncbi.nlm.nih.gov/31130514/>) that if a single plasmid goes in, the others most likely follow. However, this is indeed not always the case. To address this, we have added new data describing the efficiency of the hygromycin cassette in crisp engineering. (supplemental Figure 2)

4. Related to the previous point. Did the authors perform any analysis that showed the hygro selection contributed to the enrichment of base-edited cells?

A: The reviewer raises an important point, which we think was overlooked by us in the original submission. Electroporating the hygromycin cassette indeed works as a marker to select mutant organoids. We thought it was necessary to address this comment, so we designed the following experiment, which is presented in the revised manuscript. We transfected P53 sgRNA with and without the hygromycin resistance cassette, and compared the number of clones that successfully grew with or without hygromycin selection in Nutlin-supplemented medium, a chemical that positively selects for P53 mutant organoids. Indeed, we found that while 100% of the nutlin-3 resistant bulk culture had the TP53-W53 mutation incorporated, 72% of the hygromycin-resistant clones harbored the mutation. This was in stark contrast to the unselected and untransfected groups, which had no TP53-W53* mutation. We report this data in supplementary figure 2 of the revised manuscript.*

5. Fig 3a-e, are they cherry-picked examples? These panels can be grouped just as “panel a.” Similarly in Fig 3f,g, how many organoids are analyzed here? Are they also arbitrary picked results or population averages of many?

A: Panels 3A-E were representative images of organoids expanded from two different clones for each genotype that were stained for beta-catenin. Supplementary figure 2 shows organoids from a different clone imaged for beta-catenin expression. The shaded region surrounding the line curves in panels 3F-G represents the standard error of the mean. This information was unfortunately overlooked in the original submission. We have now clarified this in the text and have added more information in the materials and methods section. As the reviewer might notice, figures 2 and 3 (figure 1 and 2 in the current version) have been significantly revised and data on the beta-catenin mutations has been considerably expanded.

As for the reviewer’s comment about combining 5 different pieces of data into a single panel, we believe that the ability to reference individual images in the text is crucial for the reader to follow the manuscript. Moreover, panels B-E have now been moved to figure 3 to further simplify the way this section of the manuscript is presented. While we understand why the reviewer thinks these panels can be grouped into one, we believe that this is a matter of aesthetics. Therefore, we have decided to keep individual labels for the four panels in revised figure 2.

1. Cancer cells are known to be heterogeneous even though they have the same genotype. When the authors analyzed the CTNNB1 cellular localization harboring the mutation alleles, did they observe any clone-to-clone differences in CTNNB1 localization?

A: Again, the reviewer raises a very interesting and important point. Indeed, cancer cells are known to be heterogeneous regardless of their genotype. This is exactly what we see when we quantify the localization of CTNNB1 on the membrane and in the nucleus. The standard error of the mean for the quantification has a large spread, which strengthens the rationale behind our quantification approach. In the revised manuscript, we have decided to further transcriptionally characterize the difference between two genotypes generated in hepatic organoids, S45P and D32G. We posited that comparison of the effects of these two mutations on Wnt-pathway activation would be particularly interesting given the stark difference in the number of cancer cases that have been reported that harbor these mutations. To simplify the rest of the figure, quantification of beta-catenin in T41A and S33F backgrounds has now been moved to the supplementary figures.

7. Fig 1, Fig 4a, and Fig 6a do not show any new knowledge and can be omitted.

A: We moved Figure 1 and Figure 4a to the supplements (Supplemental figure 1). Readers that are interested in tumor modelling but do not know what comprises a base editor now have the opportunity to find the characteristics in the supplements.

We have decided to keep figure 6a (now figure 5A) in the revised manuscript. Once again, while we understand why the reviewer thinks this panel can be omitted, we believe that this is a matter of aesthetics. This panel allows the reader to understand why combining ABEs and CBEs in certain contexts does not work because of the mutations generated as a result of the sgRNA working with both ABEs and CBEs. This was a strong motivation for us to explore other base editors, primarily saCas9 CBE, which has been discussed in figure 6 (Now figure 5). Panel 6a (now figure 5a) also allows the reader to easily comprehend the dramatically different PAM requirements for the two Cas9 variants we are describing. Overall, we believe that keeping this panel in the figure is more advantageous for the readers. Additionally,

8. Fig 3i: Needs statistical tests.

A: We thank the reviewer for this comment. As the reviewer may appreciate in the revised version of this manuscript, we have replaced the qPCR data with RNA-sequencing of WT, S45P, and D32G mutants (2 clones each). We believe that adding this RNA-seq experiment significantly improves this section of the manuscript.

9. In Fig 4d: how were the mixed signals from heterozygous mutations treated here?

A: We intentionally chose to work only with the homozygous Q245* mutant in an effort to mimic the complete loss of PTEN. This is often observed in later stages of endometrial cancer progression. We consequently cryopreserved the heterozygous clones which remain available for further investigation. As for the R130*, unfortunately, this represents a limitation of base editing. The only sgRNA that we could design was not efficient enough to induce homozygous mutations at the R130 locus. This has now been discussed in the revised manuscript.

10. In Fig 4g,h,j: Need statistical tests.

A: Similar to our response to one of this reviewer's previous comments, we have replaced these panels with RNA-sequencing of WT, PTEN, and PTEN/PI3K double mutant endometrial organoids. We now present a heatmap in supplementary figure 2g which shows a selected list of genes with differential gene expression (positive and negative target genes of the PI3K pathway). Also, to further consider global transcriptomic changes associated with PTEN/PIK3CA mutations in endometrial organoids, we present in supplementary figure 2h, a Gene Set Enrichment Analysis (GSEA) plot showing upregulation of mTORC1 signaling in mutant organoids.

11. Fig 6b: please show all the gRNA datasets.

A: We have now included these data in the main figure. (Figure 5b)

12. Fig 6f: Does the one with TP53 mutation have any growth phenotype?

A: P53 mutant organoids continue to divide and when kept in culture without passaging, they become extremely large in size while maintaining a healthy phenotype. This is contrary to wild-type organoids that normally collapse and die once they increase too much in size. We posited that this was a result of increased proliferation of TP53 mutant cells, resulting in faster growth, especially in selection medium containing Nutlin-3. We now explain this in the text.

Minor comments

13. Line 71 “As base editors mediate genetic changes without the need for deleterious DSBs”: they often induce deletions.

A: A side-effect of base editors may indeed be (in a small percentage of edited alleles) an indel due to the nickase activity of the Cas9. However, contrary to “conventional” CRISPR-HDR to install point mutations, a DSB is not required. Thus, the indel percentage is significantly lower in base edited cells compared to cell edited via conventional HDR strategies.

14. Line 98 “To model hepatocellular tumorigenesis in human hepatocyte organoids, we designed, three sgRNAs to introduce oncogenic mutations in the CTNNB1 locus”: It is better to clarify that the genome-editing reagents were introduced to the organoids but not to ASCs.

A: Organoids used in this study were derived from adult stem cells. When we transfect organoids, they are first dissociated into a single cell suspension, with each cell holding the capacity to form a clone. Therefore, there is no distinction between organoids and ASCs, particularly within the scope of this study. To ensure that this remains clear to the audience of the manuscript, we have now added an explanation pertaining to how electroporations are performed in the materials and methods section. However, we want to clarify that this electroporation strategy has been previously described by our lab and others (for example, see Fujii, M., Matano, M., Nanki, K. & Sato, T. Efficient genetic engineering of human intestinal organoids using electroporation. Nat. Protoc. 10, 1474–1485; 2015).

15. Line 164 “We first used SpCas9-CBE to target the Q245 residue in exon 7 of PTEN”: Was this also to stem cells or to the organoids?

A: As mentioned in our response to this reviewer’s previous comment, when organoids are maintained in expansion medium (i.e. medium designed to promote the indefinite propagation of organoids), they retain their proliferative stem cell state. To introduce genetic mutations using our multiplexed ABE/CBE-based approach, we first have to recover the organoids from the 3D matrix for their dissociation into single adult stem cells. This means that we electroporate a homogeneous pool of single cells (typically in the order of 1×10^6 cells) and allow them to recover in growth medium supplemented with Rock inhibitor. The difference between stem cells and organoids is merely that of semantics (see above). In the materials and methods section we detail the electroporation protocol and we explain that the source material is a pool of single adult stem cells obtained by organoid dissociation with TrypLE.

16. Line 94 “This is followed by a GSK3-mediated phosphorylation cascade that starts at threonine-41 (T41) and the subsequent phosphorylation of serines at position 33 and 37 (S33 and S37)”: Need a reference.

A: We have added a reference supporting this statement.

17. Fig 2d-g should be combined into a single panel. Please specify the gRNA spacers and their expected target windows.

A: We believe that the ability to reference individual images in the text is crucial for the reader to follow the manuscript. While we understand why the reviewer thinks these panels can be grouped into one, we believe that this is a matter of aesthetics. Therefore, we have decided to keep individual labels for the four panels in revised figure 2. (Figure 1 in the current version)

18. Fig 3a-e should be combined into a single panel.

A: Please refer to our response to your previous comment.

19. Fig 4a is unnecessary but the codon conversation table is good to keep.

A: We have moved Figure 4a to Supplemental Figure 1, together with the CBE and ABE schematics.

20. Fig 4b: x- and y-axis labels are not aligned

A: This has been fixed.

21. Fig 4h-j: Maybe it is better to unify the color codes of the bar charts.

A: These panels have been replaced by RNA-sequencing data, see supplementary figure 2g,h. We have performed RNA-sequencing analysis and introduced a heatmap of selected genes reflecting alterations of the PI3K pathway, as well as a GSEA plot which shows upregulation of mTORC1 signaling in mutant organoids.

22. Fig 7b and c are the same data.

A: While it is true that panels B and C in figure 7 use the same data, they represent two different points of the paper. In figure 7B, the clones are aligned from cystic to non-cystic morphology (top to bottom), information that would otherwise be lost if only panel C was looked at. We have now clarified this in the text.

23. Fig 7d: Please use bar chart instead of line chart.

A: We have made this change.

24. Fig 1a: Although I think this figure does not need to be presented, the PAM is not indicated in the diagram.

A: Figure 1A represents generic Cas9 CBEs and ABEs, both of which identify the same NGG PAM. This is now better explained in the text.

25. Line 467-468: Include the voltage information for the electroporation.

A: This information is now present in the materials and methods section.

Reviewer #3 (Remarks to the Author):

Geurts, Boretto, Shashank Gandhi and colleagues present the application of base editors for the generation of tumor models in human ASC-derived organoids from multiple tissues. They show the efficacy of both cytosine and adenine base editors and use them to model hot-spot point mutations in CTNNB1, PTEN and PIK3CA APC and TP53. Moreover, they model colorectal cancer tumorigenesis by targeting 4 cancer genes in a single experiment.

This is an interesting study that reports the utility of base editors as a technology to model mutations in driver genes in ASC-derived organoids and generate biobanks of organoids harboring combinations of mutations observed in specific tumor types. The methods and the conclusions will interest cancer biologists. However, to solidify the value of their study, the authors should address few concerns.

Major

1) As stated by the authors, undesired byproducts of base editing, occur at the target site. This happens because the base editors can generate bystander edits when multiple C or A nucleotides are present within the 4-8 nucleotides editing window. In addition to on-site byproducts, undesired deaminations

at off-target DNA sites can also occur (in both Cas9-dependent and independent manner). Have the authors performed any assay (for example CIRCLE-seq) for detecting the extent of such off-targets?

A: We agree with the referee that it is important to assess off-target effects of genome engineering. As off-target detection strategies such as CIRCLE-seq and GUIDE-seq require nuclease active cas9's for their function we had to resort to whole genome sequencing. This allowed us to detect both sgRNA dependent and sgRNA independent (genome wide) off-target effects. Additionally, to further strengthen our data, we compared multiplexed versus sequentially engineered organoid clones. The data is now given in figure 7 and is further discussed in the current version of the manuscript.

2) Hepatocyte organoid outgrowth upon Nutlin-3 selection on 3 SaKKH-sgRNAs (W146*, Q165* and Q317*) was compared to a scrambled control sgRNA and to the optimal TP53-W53* sgRNA in combination with SpCas9-CBE. Is the difference in growth due to the performance of the SpCas9-CBE vs SaKKHCas9-CBE? Since the authors are comparing different nucleotides (that can suffer from chromatin accessibility effect) it's hard to distinguish between the Cas9 performance vs local effect.

A: We agree with the referee that the difference between TP53-W53 and the SaCas9 sgRNA's is remarkable. It remains difficult to predict the efficiency of a sgRNA prior to using it in a cell or organoid line. In the current version of the manuscript, we also show the data regarding the SaCas9 sgRNA that targets the Q52* mutation. This sgRNA did not result in any growing hepatocyte clones. As this mutation is very close to the W53* mutation, we believe that the difference in editing is not due to chromatin accessibility. The exact variables that determine their efficiency are not fully understood. Hence, we typically test multiple sgRNA's to achieve our knock outs of interest in TP53 and APC.*

3) Limitations of the Study: Please expand the discussion regarding the limitations of using the base editor system as a tool to generate bio banks of ASC-derived organoids. As stated by the authors, this approach doesn't allow to reproduce the 38% of transversion. Which is the percentage of mutations occurring in oncogenes and tumor suppressors that is actually possible to model with this approach?

A: We performed this exact analysis and show the data in Figure 7 of the current version of the manuscript. We analyzed all pathogenic mutations observed in oncogenes and tumor suppressors according to the COSMIC and ClinVar databases. We hope to satisfy the reviewer with this analysis.

4) Since the engineered organoids represent an excellent model to a better understand the tumorigenesis and develop now therapeutic regimens, it would be desirable to compare the transcriptome profile of the organoid models for colorectal tumorigenesis with patients-derived organoids with matching mutational profile.

A: We agree that it would be interesting to directly compare the transcriptome profiles of genetically engineered organoids to their genetic counterparts in organoid biobanks. Similar efforts have been done by others (<https://pubmed.ncbi.nlm.nih.gov/25706875/>) and thus we decided not to do it in the present study. We performed more in depth biological validation in the hepatocyte and endometrial models as these are more novel. The multiplexed colorectal organoids were used to assess off-target effects.

5) Could the authors comment how their results compare to other papers with a similar approach (for example PMID: 29969439)?

A: We would like to refer to our answer to question 2 of referee 2. We have pasted that answer below. "We thank the reviewer for this comment and for pointing out the work of our colleagues from the Dow lab. We do not refute that other labs have shown the efficiency of base editors in organoids. However, the scale of our study is significantly different. As the reviewer might know, creating cancer models that faithfully recapitulate the genetics of human cancer has been quite challenging, especially when genes harbor specific gain/loss-of-function mutations at multiple loci. Sequential introduction of such tumorigenic mutations is laborious and can sometimes take several months to obtain mutant organoids, a major limitation that was holding the field back. Our work addresses this limitation. By creating complex combinations of the exact changes associated with cancers in single transfection reactions, we demonstrate the versatility and flexibility of multiplexed CRISPR base-editing for cancer modeling, importantly across several human epithelial tissues, including endometrium, colon, and liver organoids.

We envision that our approach can be readily adapted to create in vitro models for tumorigenesis of solid human tumors for multiple tissues. We have addressed this in the introduction and the discussion in the current version of the manuscript."

Minor

1) Figure 2i. In the text is stated that all the mutant organoids were tested for the expression of Wnt-pathway genes. Since the error bars are not shown, it's not clear the variability between the mutants. Or only one mutant organoid is represented? Please specify

A: In the first version of the manuscript we indeed tested the expression of wnt-pathway genes in a single organoid clone. In the present version of the manuscript we have performed bulk RNA-sequencing to provide further backing to this data.

2) Figure 4b. The numbers are not clear

A: The axis in figure 4b has been corrected and the percentages of PTEN mutant samples are now visible. Also, since we removed Figure 4a in agreement with previous comments, Figure 4b has now been moved as Figure 4a.

3) Supplementary Figure 2a. Please correct the caption "brightfield" (overlapping with anchor)

A: This is corrected in the current version of the manuscript.

4) Supplementary Figure 2d. Specify in figure legend WT, HM, HT. axis missing.

A: We have corrected the figure legend which now describes without abbreviations each genotype. Also we corrected the X-Axis labels so that now they match the labels from Supplementary figure 2c.

5) Supplementary Figure 2g,h, j. Y axis missing

A: In agreement with a previous comment from another reviewer, we decided to replace the heatmaps in Supplementary figure 2g,h,j with dose-response curve to help better visualize the differences in response to each inhibitor. The new figures are now oriented as such; 4g-h show the dose-response curve of the mTOR inhibition through Everolimus treatment (Fig. 4g) and the bar graph reporting the IC50 of the inhibitor (Fig. 4h). Also, our new Supplementary Figures 2i-l show the dose-response curves to AZD-4547 (Supplementary Fig. 2i) and IC50 (Supplementary Fig. 2j), the dose-response curve to Alpelisib (Supplementary Fig. 2k) and IC50 (Supplementary Fig. 2l). It is now possible to see (on the Y-Axis) the viability of the samples and on the X-Axis the concentration of the inhibitor in logarithmic scale.

6) Figure 4h – "Bar graphs showing decreased gene expression of PTEN". The downregulation of PTEN is not clear in the Q130* mutant

A: The R130 mutation was only obtained on 1 allele making it a heterozygous mutant line. Therefore, we only observed a modest downregulation of gene expression while a more robust downregulation was observed with the Q245* (homozygous) mutant clones due to non-sense mediated decay of the mRNA. To avoid confusion and to meet the comments of multiple reviewers, we now included novel data obtained through RNA-sequencing and replaced the gene expression bar graphs (Figure 4h-l, Supplementary 2f) with data generated from this novel analysis, see supplementary figure 2g,h. We have performed RNA-sequencing analysis (on WT, PTENQ245* and PTENQ245*/PIK3CAE545K) and introduced a heatmap of selected genes reflecting alterations of the PI3K pathway (among them the downregulation of PTEN gene expression is evident), as well as a GSEA plot which shows upregulation of mTORC1 signaling in mutant organoids.*

7) Supplementary figure 3f: Did the authors mean "TP53 W54* sgRNA" instead of "scramble sgRNA"?

A: This is now adjusted in the current version of the manuscript.

8) What is the overall % of successfully electroporated ASCs in the different tissues?

A: While we did not directly measure the percentage of successfully electroporated ASCs, we observed GFP signal 24 hours post transfection in less than 10% cells across the three different organoid lines that were used in this study. The percentage of mutant clones that were sequenced and their genotype confirmed by Sanger sequencing has been mentioned throughout the manuscript.

9) Capital letters in figure legends are not consistent. Please, correct.

A: This is now adjusted in the current version of the manuscript.

REVIEWERS' COMMENTS

Reviewer #1 (Remarks to the Author):

The authors made much clearer explanations and additional experiments to alleviate most of my concerns. Here are some remaining minor issues.

Re to : We have also removed panel 'K' from figure 4 and replaced the heatmaps with dose-response curves, as suggested by this reviewer. We have also replaced the bar graphs showing the AUC with ones that show the IC50 for the inhibitor in each genetic background. Following suggestions by the reviewers, the revised figures 4g-h now show the dose-response curve of the mTOR inhibition through Everolimus treatment (Fig. 4g) and the bar graph reporting the IC50 of the inhibitor (Fig. 4h). Also, the revised Supplementary Figure 2i-l shows the dose-response curves to AZD-4547 (Supplementary Fig. 2i) and IC50 (Supplementary Fig. 2j), the dose-response curve to Alpelisib (Supplementary Fig. 2k) and IC50 (Supplementary Fig. 2l). We believe that the effect of these mutations on lowering the sensitivity of the organoids to different PI3K pathway inhibitors is thus clearly evident. Finally, we acknowledge the reviewer's comments about figure 7.

This is a great addition (though the Figure call out is wrong in this response) but not without methodological details. Please update how the fitting was performed and IC50 was determined. Only presenting the graph and brief legend are inadequate.

Re to: "According to our database-research the T41A mutation is caused by c.121A>G (<https://www.ncbi.nlm.nih.gov/clinvar/RCV000019143/>)"

What I meant was your graph error. In new Fig1b, you pointed c124A>G with gray box. However, in new Fig1e, c.121A>G is indicated. This is very puzzling and correct either fig to be accurate.

Re to : "A: In this manuscript, we have tested 8 different concentrations of 3 inhibitors on 3 different genetic backgrounds (WT, PTENQ245*, and PTENQ245*/PIK3CAE545K), which we

believe constitutes a drug screen. Yet, we have reworded this and now use the term drug sensitivity testing. “

There is still wording around screening and it needs to be corrected to drug sensitivity testing. :

Sentence1 "By performing drug screening assays on organoids harboring either PTEN or PIK3CA mutations, we unravel the mechanism underlying the initial stages of endometrial tumorigenesis.", Sentence2 "Drug screening on endometrial organoids Drug screenings were performed as previously described 71. In brief, the organoids were recovered from BME and enzymatically dissociated in TrypLE for 3 minutes at 37°C....."

Also new typos are found, such as "Inspite" "transcriptional", "Interestinly". Please carefully address these before being published.

Reviewer #2 (Remarks to the Author):

I think the authors greatly addressed most of the reviewers' comments and improved the manuscript. Below, I list my remaining comments:

1. Missing scale bars in figures: for example, Fig. 3 and Supplementary Fig. 4. Please check this throughout the manuscript.
2. Line 225 "that while 178 genes were differentially enriched in the 226 D32G mutant, 185 genes were differentially enriched in the S45P background" - please also discuss about depleted genes and perform GO analyses.
3. In response to my previous comment, "22. Fig 7b and c are the same data", they claimed "In figure 7B, the clones are aligned from cystic to non-cystic morphology (top to bottom)", but how readers can see the information of cystic and non-cystic clones. please have an additional column to indicate cystic/non-cystic morphotypes.
4. Supplementary Fig. 2a: Is this a bright field image? No channel annotation
5. Supplementary Fig. 3b: No fluorescent channel name

6. Supplementary Fig. 4d: no x- and y-axis labels

7. Supplementary Fig. 4f: x-axis label is flipped and does not match to other panels.

Reviewer #3 (Remarks to the Author):

The authors have addressed my concerns.

Reviewer #1 (Remarks to the Author):

The authors made much clearer explanations and additional experiments to alleviate most of my concerns. Here are some remaining minor issues.

Re to : We have also removed panel 'K' from figure 4 and replaced the heatmaps with dose-response curves, as suggested by this reviewer. We have also replaced the bar graphs showing the AUC with ones that show the IC50 for the inhibitor in each genetic background. Following suggestions by the reviewers, the revised figures 4g-h now show the dose-response curve of the mTOR inhibition through Everolimus treatment (Fig. 4g) and the bar graph reporting the IC50 of the inhibitor (Fig. 4h). Also, the revised Supplementary Figure 2i-l shows the dose-response curves to AZD-4547 (Supplementary Fig. 2i) and IC50 (Supplementary Fig. 2j), the dose-response curve to Alpelisib (Supplementary Fig. 2k) and IC50 (Supplementary Fig. 2l). We believe that the effect of these mutations on lowering the sensitivity of the organoids to different PI3K pathway inhibitors is thus clearly evident. Finally, we acknowledge the reviewer's comments about figure 7.

This is a great addition (though the Figure call out is wrong in this response) but not without methodological details. Please update how the fitting was performed and IC50 was determined. Only presenting the graph and brief legend are inadequate.

A: We thank the reviewer for pointing out the mistakes in the figure call out and have checked all figure call-outs throughout the manuscript. Indeed we did not specify the methodology for IC50 determination thus we have included this in the current version of the manuscript in the figure legend and the materials and methods. In short: The dose-response curves and IC50 values were generated with GraphPad Prism v7.03 using the nonlinear regression model (least squares fitting method).

Re to: "According to our database-research the T41A mutation is caused by c.121A>G (<https://www.ncbi.nlm.nih.gov/clinvar/RCV000019143/>)" What I meant was your graph error. In new Fig1b, you pointed c124A>G with gray box. However, in new Fig1e, c.121A>G is indicated. This is very puzzling and correct either fig to be accurate.

A: We have ensured that both instances are now c.121A>G.

Re to : "A: In this manuscript, we have tested 8 different concentrations of 3 inhibitors on 3 different genetic backgrounds (WT, PTENQ245*, and PTENQ245*/PIK3CAE545K), which we believe constitutes a drug screen. Yet, we have reworded this and now use the term drug sensitivity testing. "There is still wording around screening and it needs to be corrected to drug sensitivity testing. Sentence1 "By performing drug screening assays on organoids harboring either PTEN or PTEN and PIK3CA mutations, we unravel the mechanism underlying the initial stages of endometrial tumorigenesis.", Sentence2 "Drug screening on endometrial organoids Drug screenings were performed as previously described 71. In brief, the organoids were recovered from BME and enzymatically dissociated in TrypLE for 3 minutes at 37°C....."

A: We indeed did not change the wording regarding drug screening in the abstract and methods section of the previous version of the manuscript. This is now addressed and changed to drug sensitivity testing/assays.

Also new typos are found, such as "Inspite" "transcriptional", "Interestinly". Please carefully address these before being published.

A: We went carefully over the manuscript and have fixed the typos that were introduced during the revision process.

Reviewer #2 (Remarks to the Author):

I think the authors greatly addressed most of the reviewers' comments and improved the manuscript. Below, I list my remaining comments:

1. Missing scale bars in figures: for example, Fig. 3 and Supplementary Fig. 4. Please check this throughout the manuscript.

A: All figures are now replotted according to Nature's guidelines regarding figures and data visualization.

2. Line 225 "that while 178 genes were differentially enriched in the 226 D32G mutant, 185 genes were differentially enriched in the S45P background" - please also discuss about depleted genes and perform GO analyses.

A: In our analysis we directly compare D32G to S45P, thus we discuss depleted genes in one of the genotype as upregulated genes in the other. To further strengthen our claims, we have performed GO term analysis and we indeed see a difference between the two phenotypes. This is now highlighted in the text " Gene Ontology analysis on the list of enriched genes in the D32G background revealed a significant upregulation of the "plasma membrane" cellular compartment."

3. In response to my previous comment, "22. Fig 7b and c are the same data", they claimed "In figure 7B, the clones are aligned from cystic to non-cystic morphology (top to bottom)", but how readers can see the information of cystic and non-cystic clones. Please have an additional column to indicate cystic/non-cystic morphotypes.

A: We have added a supplementary dataset that contains pictures of all clones that were sequenced in this manuscript. This now better aligns with the text and the reader is now able to assess organoid phenotypes. This is supplementary dataset 1 in the current version of the manuscript.

4. Supplementary Fig. 2a: Is this a bright field image? No channel annotation

5. Supplementary Fig. 3b: No fluorescent channel name

6. Supplementary Fig. 4d: no x- and y-axis labels

7. Supplementary Fig. 4f: x-axis label is flipped and does not match to other panels.

A: We have addressed all figure issues and have made sure that our figures now align with Nature's guidelines regarding figures and visualization of data.